# Whole-genome scanning reveals environmental selection mechanisms that shape diversity in populations of the epipelagic diatom *Chaetoceros*

Charlotte Nef [1,2], Mohammed-Amin Madoui[3,4], Éric Pelletier[2,5], Chris Bowler [1,2]*

**1** Institut de Biologie de l'École Normale Supérieure (IBENS), École Normale Supérieure, CNRS, INSERM, PSL Université Paris, Paris, France, **2** Research Federation for the study of Global Ocean Systems Ecology and Evolution, FR2022/Tara Oceans, Paris, France, **3** Service d'Etude des Prions et des Infections Atypiques (SEPIA), Institut François Jacob, Commissariat à l'Energie Atomique et aux Energies Alternatives (CEA), Université Paris Saclay, Fontenay-aux-Roses, France, **4** Équipe Écologie Évolutive, UMR CNRS 6282 BioGéoSciences, Université de Bourgogne Franche-Comté, Dijon, 21000, France, **5** Metabolic Genomics, Genoscope, Institut de Biologie François-Jacob, CEA, CNRS, Université Evry, Université Paris Saclay, Evry, France

* cbowler@biologie.ens.fr

**Data Availability Statement:** All relevant data are within the paper and its Supporting Information files, except for Supplementary Data files S6, S8,

## Abstract

Diatoms form a diverse and abundant group of photosynthetic protists that are essential players in marine ecosystems. However, the microevolutionary structure of their populations remains poorly understood, particularly in polar regions. Exploring how closely related diatoms adapt to different environments is essential given their short generation times, which may allow rapid adaptations, and their prevalence in marine regions dramatically impacted by climate change, such as the Arctic and Southern Oceans. Here, we address genetic diversity patterns in *Chaetoceros*, the most abundant diatom genus and one of the most diverse, using 11 metagenome-assembled genomes (MAGs) reconstructed from *Tara* Oceans metagenomes. Genome-resolved metagenomics on these MAGs confirmed a prevalent distribution of *Chaetoceros* in the Arctic Ocean with lower dispersal in the Pacific and Southern Oceans as well as in the Mediterranean Sea. Single-nucleotide variants identified within the different MAG populations allowed us to draw a landscape of *Chaetoceros* genetic diversity and revealed an elevated genetic structure in some Arctic Ocean populations. Gene flow patterns of closely related *Chaetoceros* populations seemed to correlate with distinct abiotic factors rather than with geographic distance. We found clear positive selection of genes involved in nutrient availability responses, in particular for iron (e.g., ISIP2a, flavodoxin), silicate, and phosphate (e.g., polyamine synthase), that were further supported by analysis of *Chaetoceros* transcriptomes. Altogether, these results highlight the importance of environmental selection in shaping diatom diversity patterns and provide new insights into their metapopulation genomics through the integration of metagenomic and environmental data.

S10 and S17 that are available from the Zenodo database (accession numbers https://doi.org/10.5281/zenodo.7189704, https://doi.org/10.5281/zenodo.7189810, https://doi.org/10.5281/zenodo.7189835).

**Funding:** This work was supported by the European Research Council (ERC) under the European Union's Horizon 2020 research and innovation programme (Diatomic; grant agreement No. 835067 to CB and CN). Additional funding is acknowledged from the French Government "Investissements d'Avenir" Programmes MEMO LIFE (Grant ANR-10-LABX-54 to CB), Université de Recherche Paris Sciences et Lettres (PSL) (Grant ANR-125311-IDEX-0001-02 to CB); and France Génomique (ANR-10-INBS-09 to CB), and OCEANOMICS (Grant ANR-11-BTBR-0008 to CB), which were funded through Agence Nationale de la Recherche. The funders had no role in study design, data collection and analysis, decision to publish, or preparation of the manuscript.

**Competing interests:** The authors have declared that no competing interests exist.

**Abbreviations:** AAI, average amino acid identity; ANI, average nucleotide identity; BAF, B-allele frequency; BUSCO, Benchmarking Universal Single-Copy Orthologs; DCM, deep-chlorophyll maximum; GO, Gene Ontology; ISIP, iron starvation-induced protein; LMM, linear mixed model; MAG, metagenome-assembled genome; MMETSP, Marine Microbial Eukaryote Transcriptome Sequencing Project; OTU, operational taxonomic unit; PCA, principal component analysis; PTM, posttranslational modification; SNV, single-nucleotide variant; SUR, surface; UTR, untranslated region.

## Introduction

About half of primary productivity on Earth is supported by aquatic phytoplankton, a phylogenetically diverse group of photosynthetic organisms composed of eukaryotic algae and cyanobacteria that provide essential ecosystem services, from nutrient cycling and $CO_2$ regulation to sustaining higher trophic levels as the base of marine food webs [1–3]https://www.zotero.org/google-docs/?pyFXQV. Among phytoplankton, diatoms are pivotal in marine ecosystems since they account for an estimated 40% marine primary productivity and 20% global carbon fixation [2], as well as being important contributors to global carbon export [4]. Moreover, they link silicon and carbon biogeochemical cycles through the synthesis of their elaborate silicified cell walls, surrounded and embedded by glycoproteins that prevent its dissolution [5,6]. Diatoms are therefore key players also in the global silicon cycle, particularly in the Southern Ocean [7,8].

Like other pelagic plankton, diatoms are thought to display high dispersion potential due to their rapid generation times and large population sizes, combined with the few apparent oceanic barriers to dispersal [9,10]. As a consequence, they are expected to show reduced diversity patterns and biogeographic structure due to homogenised genetic pools [11]. Instead, molecular surveys have revealed that diatom populations exhibit tremendous diversity, with more than 4,000 different operational taxonomic units (OTUs) [12], while being widely distributed across all major oceanic provinces [13] encompassing high latitudes, upwelling regions as well as stratified waters [14,15]. The ecological success of diatoms is undoubtedly linked to their complex evolutionary history, which was found to be sustained by horizontal gene transfers from bacteria [16,17], and mosaic plastid evolution derived from both red and green algae [18–20]. This chimeric origin led to specific physiological innovations, such as silicon utilisation for cell protection, efficient nutrient uptake systems allowing rapid responses to environmental fluctuations, a functional urea cycle, and potential carbon concentration mechanisms [16,17,21]. In contrast to these genetically encoded functions, diatom genomes themselves appear to display a wide variety of dynamics, e.g., through specific transposable elements detected in the model diatoms *Thalassiosira pseudonana* and *Phaeodactylum tricornutum* [16,22], alternative splicing [23], as well as gene copy number variation and mitotic recombination between homologous chromosomes [24]. Altogether, these characteristics likely fuel diatom diversity, leading to rapid diversification rates [17] while increasing their ability to respond to changing environmental conditions.

Climate change is expected to induce a range of environmental stressors on phytoplankton [25]. Among these are increased water temperature and stratification, nutrient paucity, and acidification [26]. Moreover, a recent study indicated that numerous important diatom genera, such as *Chaetoceros*, *Porosira*, and *Proboscia*, are predicted to be vulnerable to climate change, particularly in polar plankton communities [27]. Diatoms appear therefore to be valuable candidates to investigate the fundamental links between their genomes, physiology, and population dynamics, in light of predicted environmental changes. Understanding such principles would require access to the genome of natural diatom populations as well as precise contextual information. With the emergence of new sequencing technologies and processes to recover genomes from environmental data, either from metagenomes or single-cell genomes, it is now possible to access the genomic information of organisms by going beyond culture-dependent approaches, allowing us to gain insights into the biology and ecology of natural populations [28,29]. This is of particular interest for organisms for which culture conditions cannot be mimicked easily, as for instance organisms thriving in polar environments. These new techniques have enabled the scientific community to access sequences from taxa lacking significant information, such as Euryarchaeota [28], Picozoa [30,31], MOCHs (for Marine OCHrophytes)

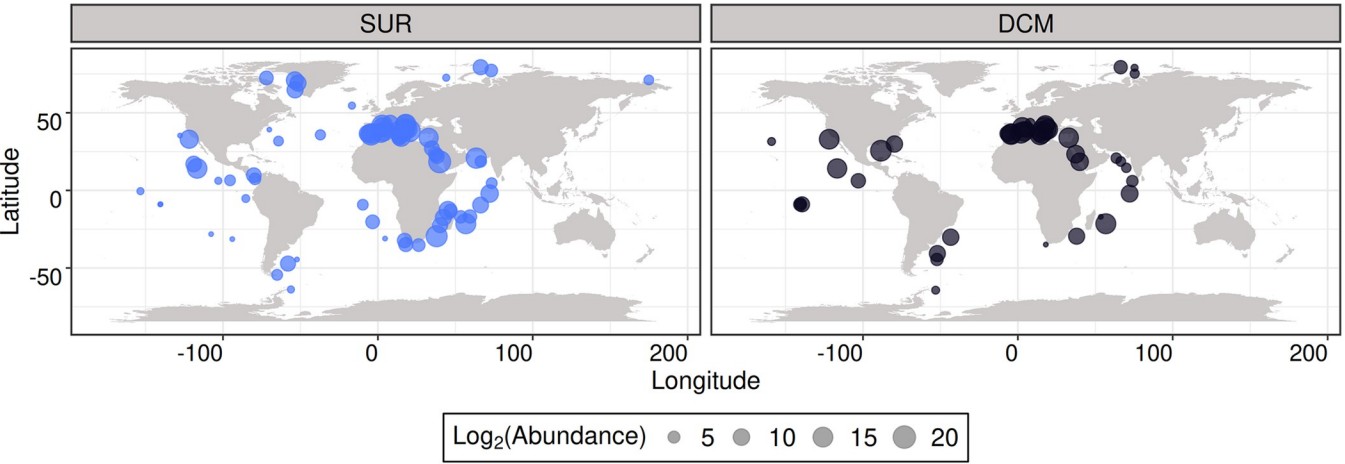

**Fig 1. Global patterns of *Chaetoceros* genus distribution based on *Tara* Oceans metabarcoding data of V9 18S rDNA.** The 4 eukaryotic-enriched size fractions 0.8–5 μm, 5–20 μm, 20–180 μm, and 180–2,000 μm were pooled for a total of $n = 2,739$ OTU barcodes in SUR and DCM depths (see S1 Data for raw values). The basemap was generated from the Natural Earth data (http://www.naturalearthdata.com/downloads/50m-physical-vectors/); Natural Earth—All maps are public domain (http://www.naturalearthdata.com/about/terms-of-use/). DCM, deep-chlorophyll maximum; OTU, operational taxonomic unit; SUR, surface.

[32], MAST-4 (for MArine STramenopiles) [29], and rappemonads [33]. Among diatoms, the genus *Chaetoceros* holds a particular position as it is the most widespread, presenting a worldwide distribution from pole to pole with a prevalence at high latitudes (Fig 1) [13,34,35].

As such, it is considered an important driver of carbon export and silica sinking in the modern ocean [4,36]. The genus displays a high level of diversity, with 239 accepted species names in Algaebase, in addition to 153 names under debate or yet to be verified (https://www.algaebase.org, as of May 2022). It is generally accepted that the *Chaetoceros* genus is subdivided into the *Hyalochaete* and *Phaeoceros* subgenera, the latter including the type species *Chaetoceros dichaeta*, though their exact subdivision remains under debate [37]. *Chaetoceros* presents peculiar physiological properties that may be responsible for its prevalent distribution. For example, some *Chaetoceros* species have been shown to display unusually high C:N ratios unaffected by light regime and nitrogen source, suggesting a capacity to accumulate superior carbon per nitrogen units than other Arctic diatoms, while showing physiological responses similar to those of more temperate diatoms [38]. Besides its particular physiological characteristics, *Chaetoceros* is known to participate in a significant range of associations with a wide variety of microorganisms. The *Chaetoceros* phycosphere has been shown to gather a diverse set of epibiotic bacteria, the composition of which simplifies along subculturing [39], and is significantly influenced by nutrient availability and host growth stage [40]. Some associated bacteria have even been observed to favour resistance of *Chaetoceros* cells against viral infection and lysis compared to axenic controls [41]. *Chaetoceros* can be involved in photosymbioses with epibiotic peritrich and tintinnid ciliates [42], interact with nitrogen-fixing cyanobacteria in diatom–diazotroph associations [43,44] and is globally highly connected with other plankton members in the *Tara* Oceans network of planktonic associations [45]. Therefore, given the ecological significance of *Chaetoceros* and its prevalence in regions particularly predicted to be vulnerable to climate change, the present study focuses on describing patterns of genetic diversity and population structure of this diatom genus. To this end, we leveraged 11 metagenome-assembled genomes (MAGs) originating from the *Tara* Oceans expeditions [46] and that are associated with highly contextualised metadata. We aimed to answer the following questions: How are natural *Chaetoceros* populations structured? Is geographic distance

a barrier to gene flow and, if not, what main ecological factors are correlated with *Chaetoceros* micro-diversification? What are the genetic functions undergoing selection among different *Chaetoceros* populations?

## Results

### Description and comparative analysis of *Chaetoceros* MAGs

We first performed a comparative analysis of 11 *Chaetoceros* MAGs (see Table A in S1 Table for details on their names) previously assembled from *Tara* Oceans metagenomic co-assemblies [46]. The MAGs globally displayed genome sizes ranging from 10.6 (ARC_232) to 44.4 (PSW_256) Mbp that are the same order of magnitude as the genomes of the model diatoms *Chaetoceros tenuissimus*, *T. pseudonana*, and *P. tricornutum* (Fig 2A).

Overall, the genomes displayed good completion percentages with 7 out of the 11 MAGs having a BUSCO score at least equal to 50% (Fig 2B, see S2 Table for the complete summary of BUSCO scores). The MAG gene numbers, ranging from approximately 5,000 to approximately 17,000 genes, adequately mirrored the genome sizes (Fig 2C). The average percentage of G+C ranged from 39% (ARC_267) to 44% (SOC_37), which is lower than those of *T. pseudonana* and *P. tricornutum*, but in line with that of *C. tenuissimus*, with a global decreasing percentage of G+C from first to third position in the codons (S1A and S1B Fig and Table D in S1 Table). Overall, *T. pseudonana* exhibited genomic characteristics closer to the MAGs and *C. tenuissimus* genome compared with *P. tricornutum*, which is consistent with *Chaetoceros* and *Thalassiosira* genera being classified as diatoms with centric symmetry. Mean gene lengths varied between 400 and 500 bp (Fig 2D), again lower than those of *T. pseudonana* and *P. tricornutum*, but expected because MAGs are generally more fragmented than genomes sequenced from cultured organisms [47]. A principal component analysis (PCA) on 8 genome and gene metrics was conducted to test whether the MAGs belonging to the same geographical region displayed common genome characteristics (Fig 2E and 2F). The PCA showed 3 defined groups: one consisting of ARC_116, ARC_217, PSE_171, PSE_253, and SOC_37 that are characterised by a larger number of genes, large genome and gene sizes as well as higher G+C content, indicating a more compact genome. A second group consisted of ARC_267 and PSW_256 that display the largest intron sizes, and a last one grouped ARC_232 and SOC_60, the smallest *Chaetoceros* MAGs. This analysis did not reveal any clustering of the MAGs based on their geographical origins. It must be noted that some of the differences observed regarding for instance genome size may be linked to the reconstruction methods applied rather than to bona fide biological differences, as exemplified by different genome completion levels (see Fig 2B).

Relatively weak average nucleotide identity (ANI) (<80%) and average amino acid identity (AAI) (<60%) were observed between the MAGs that were derived from the same geographical areas, suggesting that the populations are not necessarily closely related (Figs 3A and S2A and Sheets A and B in S2 Data). Pairwise ANI and AAI values were highly positively correlated particularly for pairwise ANI > 85% and AAI > 75% (S2A and S2B Fig). The most elevated ANI (95.1%) and AAI (94%) were observed between the MAGs ARC_217 and PSE_171. Elevated pairwise ANI (>80%) and AAI (>60%) values were also observed for PSE_253 and ARC_189, as well as for PSW_256 and ARC_267.

We then evaluated the relatedness of the *Chaetoceros* MAGs between one another and with respect to other taxa based on a concatenated tree of 34 taxa for 83 single-copy nuclear genes (total 42,525 amino acids) across the eukaryotic tree of life (Fig 3B and S3 Data). An additional tree was built with the 34 taxa as input to identify their orthogroups based on protein sequences (*n* = 846 orthogroups) (S3 Fig and S4 Data). For both analyses, we obtained a relatively good phylogeny of the taxa, with a monophyly of the diatoms. As observed previously,

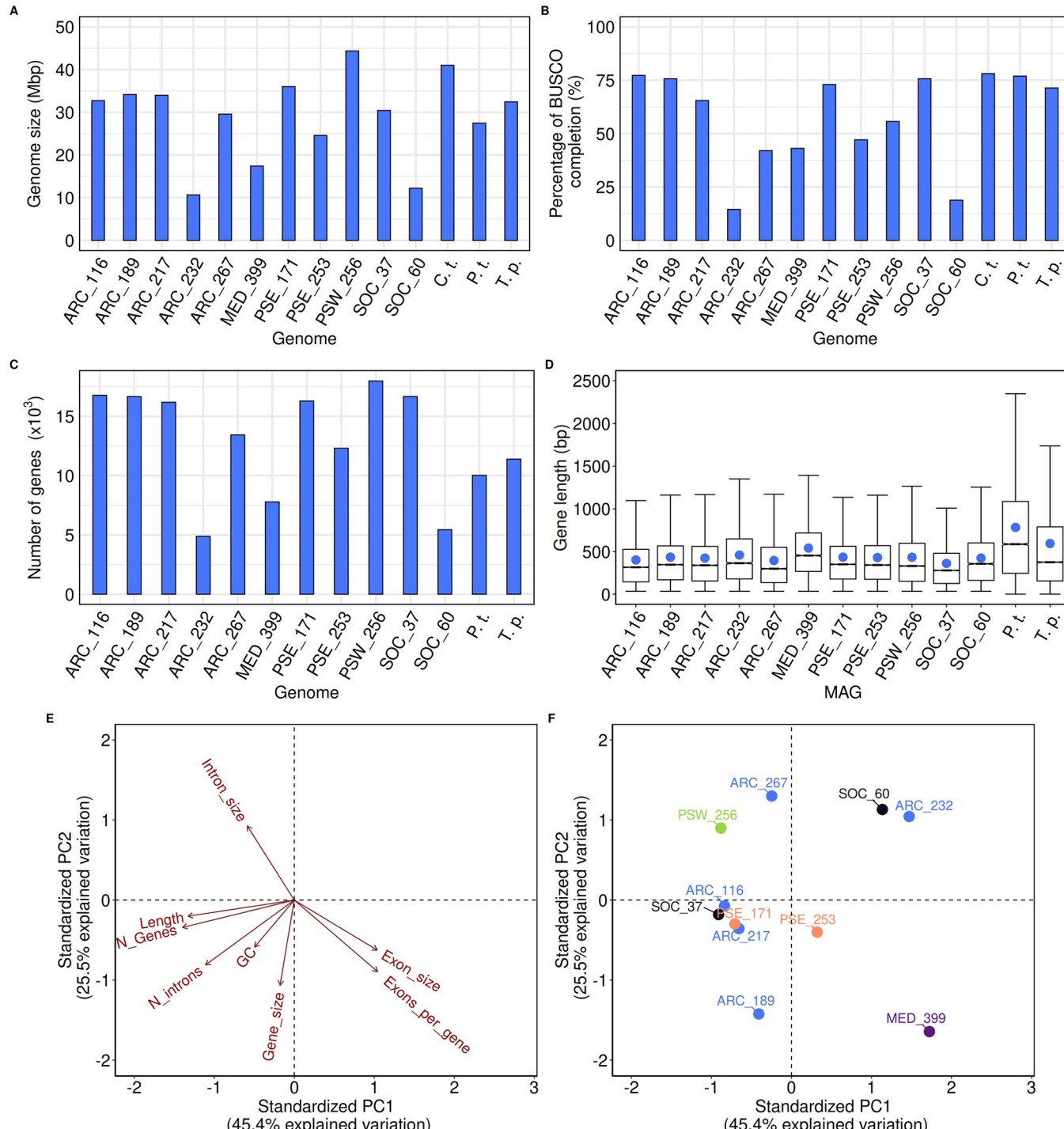

**Fig 2. Characteristics of the different *Chaetoceros* MAGs.** (A) Genome size, (B) level of BUSCO completion (*n* = 255 BUSCOs), (C) number of genes, and (D) boxplots of mean gene length (mean gene length is represented by the blue dot) of the MAGs and reference diatoms *C. tenuissimus* (C.t.), *P. tricornutum* (P.t.), and *T. pseudonana* (T.p.). Only the assembly scaffolds of *C. tenuissimus* were available, preventing us from investigating the number of genes and their length. (E, F) PCA of different gene and genome metrics of the MAGs, shaded by geographical origin (blue: Arctic Ocean; purple: Mediterranean Sea; orange: Pacific South Eastern Ocean; green: Pacific South Western Ocean; black: Southern Ocean) (See Tables B and C in S1 Table for raw values). BUSCO, Benchmarking Universal Single-Copy Orthologs; MAG, metagenome-assembled genome; PCA, principal component analysis.

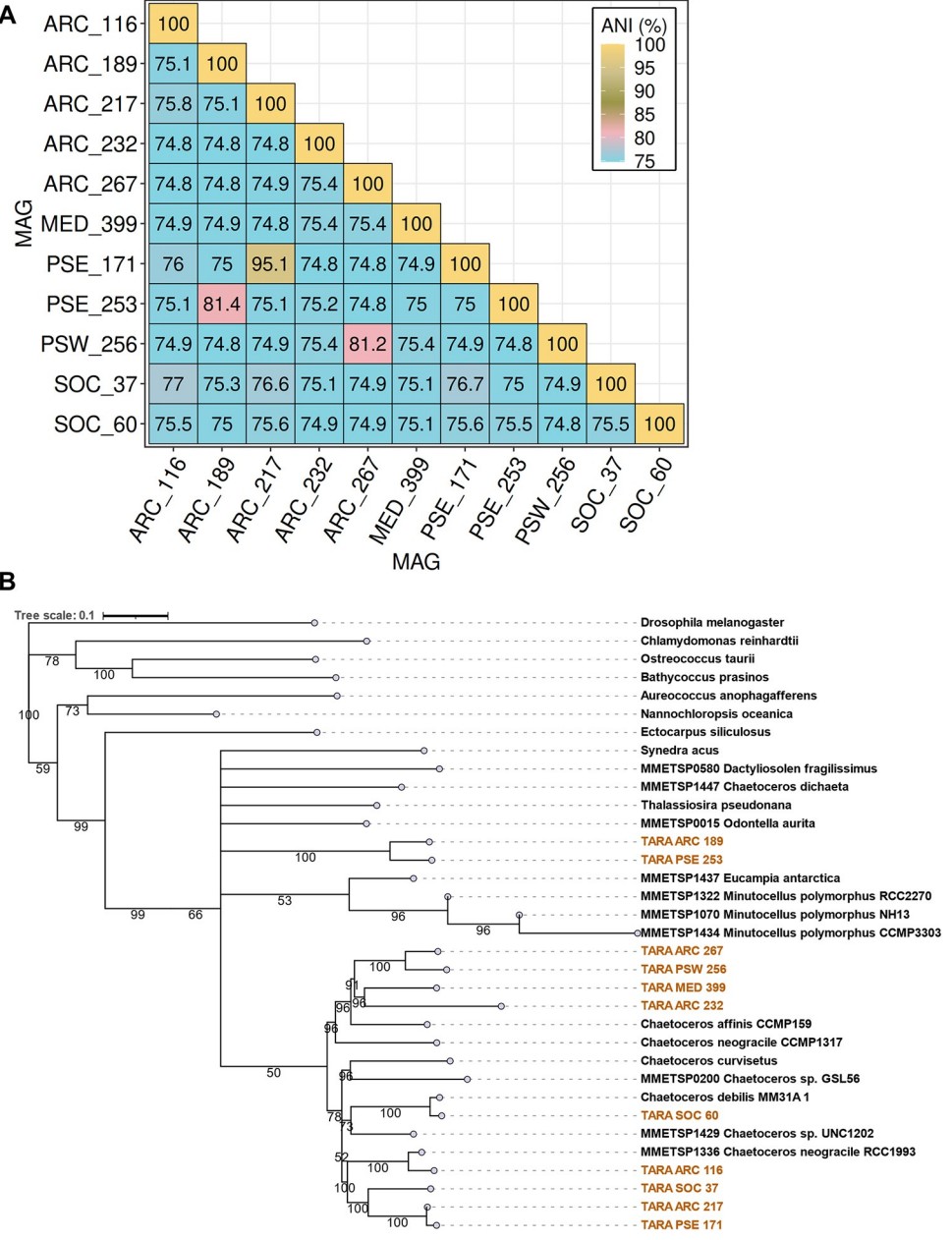

**Fig 3. Relatedness of the *Chaetoceros* MAGs.** (A) ANI. (B) Concatenated multigene ML tree generated with RAxML (100 bootstrap), based on 83 BUSCO gene clusters with the *Chaetoceros* MAGs highlighted in gold. Bootstrap values ≥ 50% are indicated. ANI, average nucleotide identity; BUSCO, Benchmarking Universal Single-Copy Orthologs; MAG, metagenome-assembled genome.

MAGs from a close geography did not appear to resolve together. In accordance with the ANI/AAI patterns, 3 MAG pairs resolved together with high support values, namely, ARC_189 and PSE 253, ARC_217 and PSE_171, and ARC_267 and PSW_256 (Figs 3B and S3). Both ARC_189 and PSE_253 MAGs resolved somewhat differently in the clade, which may suggest that they belonged to the *Phaeoceros* subgenera. Conversely, the 9 other MAGs resolved in the same clade as *C. affinis*, *C. curvisetus*, and *C. debilis*, indicating closeness to the *Hyalochaete* subgenera. The MAGs ARC_267, PSW_256, ARC_232, and MED_399 resolved in clades close

to *C. affinis* CCMP159. The MAG SOC_60 appears to be most closely related to *C. debilis* with high support (bootstrap value > 90%), while ARC_116 was closely related to the *C. neogracile* RCC1993 strain but not to *C. neogracile* CCMP1317.

Additional analyses of the MAG coding potential were conducted (see S1 Text). The shared orthologous genes revealed particular closeness of ARC_217 and PSE_171 MAGs (S4A Fig and Sheet A in S5 Data). Clear discrepancies were observed regarding the proportions of amino acids encoded in the MAGs with 2 distinct clusters, suggesting a replacement of some residues displaying the same chemical properties (S4B Fig and Sheet B in S5 Data). We further conducted a comparative analysis at the level of PFAM domains to test the relative genome enrichment in putative biological functions. No clear distinction between the MAGs based on their geographical localisation was observed (S4C and S5 Figs and Sheet C in S5 Data and S3 Table).

## Genome-resolved biogeography of *Chaetoceros*

The biogeographical distribution of the MAGs was investigated by estimating the proportion of *Tara* Oceans metagenomic reads that mapped to the 11 genomes. After filtration of the reads based on their identity and coverage (see Materials and methods, data available in https://doi.org/10.5281/zenodo.7189704, and S6 and S7 Figs for details), a final number of 20 different sampling stations and/or depths were conserved. The *Chaetoceros* MAGs together recruited 0.71% of reads from these stations (all size fractions combined) and presented an amphitropical distribution, with a large prevalence in the Arctic Ocean and minor dispersal in the Pacific and Southern Oceans, as well as in the Mediterranean Sea. The relative contribution of the MAGs to the total metagenomic reads ranged from 0.03% for SOC_60 in the Southern Ocean to a local maximum of up to 4% for ARC_116 in the Arctic Ocean (Tables A and B in S4 Table). For a given depth, some stations (i.e., TARA_92, TARA_173, TARA_188, TARA_189, TARA_194, TARA_201, and TARA_205) appeared to harbour 2 and up to 3 different *Chaetoceros* MAGs (Fig 4A), which suggests that our approach was precise enough to discriminate a mixture of populations from strains that are expected to be closely related.

The 4 MAGs ARC_116, ARC_217, ARC_232, and SOC_37 were retrieved at both the surface and the deep-chlorophyll maximum (DCM) of the water column, with ARC_116 being the most widespread and dominant *Chaetoceros* MAG at the surface and SOC_37 at the DCM. The 6 MAGs ARC_189, ARC_267, PSE_171, PSE_253, PSW_256, and SOC_60 were found only at the surface of the water column while MED_399 was the only one retrieved solely at the DCM. Of note, SOC_37 was expected to be associated with both the Arctic and Southern Oceans (see Supporting information Table S5 in [46]), but it appeared to be restricted only to the Arctic Ocean in our analysis, likely due to the stringency of our filtration parameters. The co-occurrence patterns of the MAGs were addressed by performing pairwise correlation tests between the 11 MAGs using their metagenomic abundance. It appeared that none of the MAGs displayed significant co-occurrences or mutual exclusions, except for MAGs PSE_171 and PSE_253 ($p$-value < 0.05) that were associated with the same unique station (TARA_92) (Fig 4B and Sheets A and B in S6 Data). Each of the different MAGs appeared restricted to a distinct environment characterised by a narrow range of temperature between 1 and 4°C for the same MAG (Fig 4C and Sheet C in S6 Data). Conversely, most of the MAG populations appeared to be distributed across a rather large spectrum of iron, silicate, phosphate, and nitrate concentrations (S8 Fig and Sheet C in S6 Data).

## Investigating genomic differentiation between *Chaetoceros* MAGs

***Chaetoceros* SNV landscape.** We then examined the level of genomic diversity in the different *Chaetoceros* populations by identifying for each MAG their respective single-nucleotide

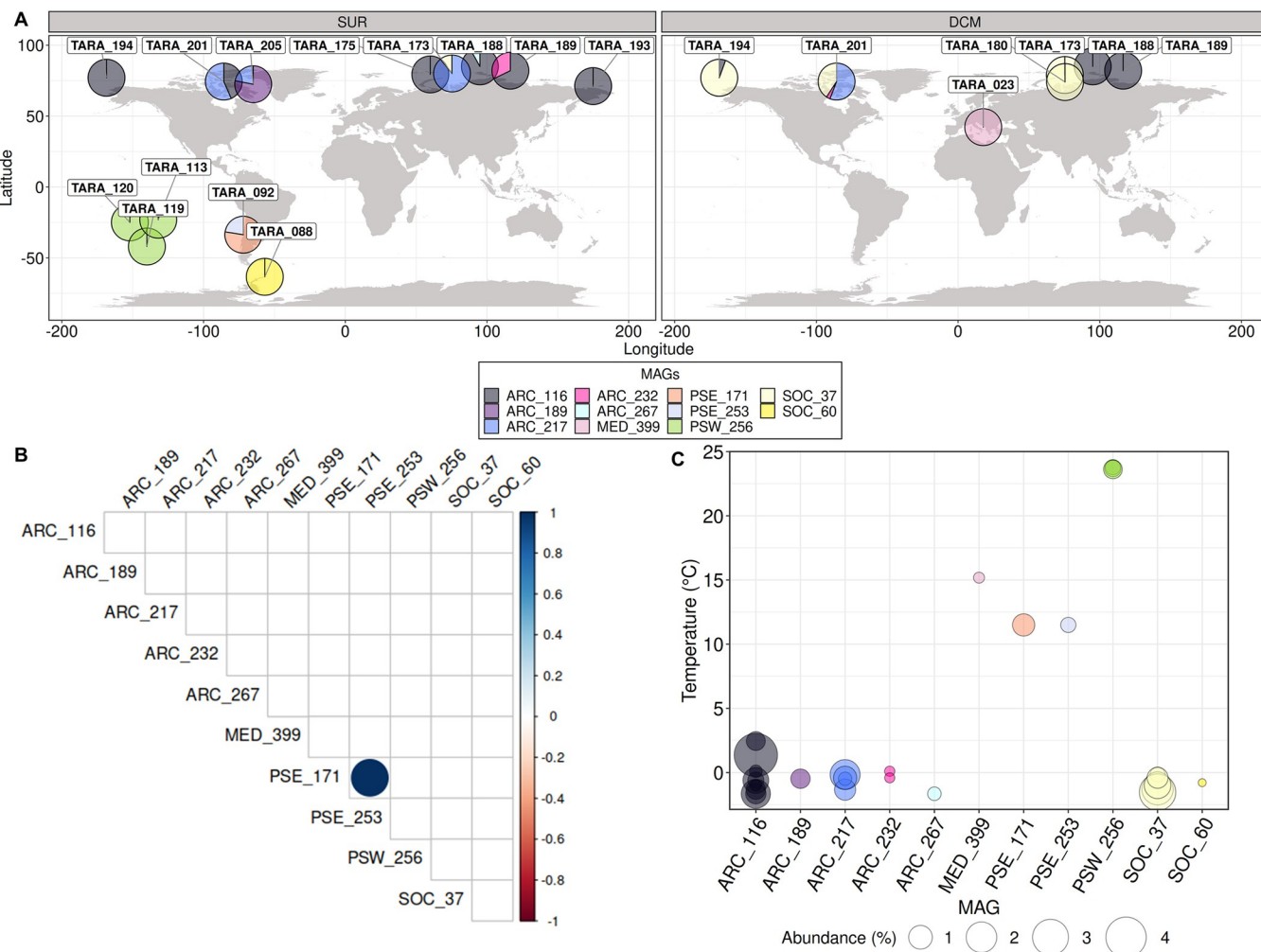

**Fig 4. Biogeography of the *Chaetoceros* MAGs throughout *Tara* Oceans sampling sites.** (A) Relative contributions of the *Chaetoceros* MAGs in SUR and DCM depths (see Tables A and B in S4 Table for details). The basemap was generated from the Natural Earth data (http://www.naturalearthdata.com/downloads/50m-physical-vectors/); Natural Earth—All maps are public domain (http://www.naturalearthdata.com/about/terms-of-use/). (B) Pairwise correlation patterns of the MAG relative abundances across *Tara* Oceans stations (the colour bar represents Spearman's correlation *rho*; values are shown when *p*-value is inferior to 0.05). (C) Bubble plot corresponding to the measured temperature at the sampling stations where each MAG was detected. DCM, deep-chlorophyll maximum; MAG, metagenome-assembled genome; SUR, surface.

variants (SNVs). A total of 8,425,600 variants were recruited, ranging from 100,000 to approximately 800,000. Globally, no significant correlation between the genome coverage and number of variants retrieved was observed (Pearson's correlation *rho* = −0.22; *p*-value = 0.26) (Fig 5A and Table A in S5 Table). Some MAGs, such as ARC_217, ARC_232, and PSW_256 nonetheless displayed variant patterns that followed the number of reads. Consequently, we assumed that the number of variants did not necessarily follow genome coverage and was rather dependent on the genome considered.

The highest local SNV level ranged from 0.63% for ARC_217 at station TARA_205 (SUR) to as much as 2.34%, observed for ARC_116, which exhibited the highest range in terms of SNV levels, at station TARA_194 (DCM) in the Arctic Ocean (Figs 5B and S9 and Table A in S5 Table). This suggests that the ANI of each MAG population to its respective consensus genome ranged up to about approximately 98% a strong indicator illustrating the occurrence of local species harbouring non-negligible micro-diversity traits in different populations. We

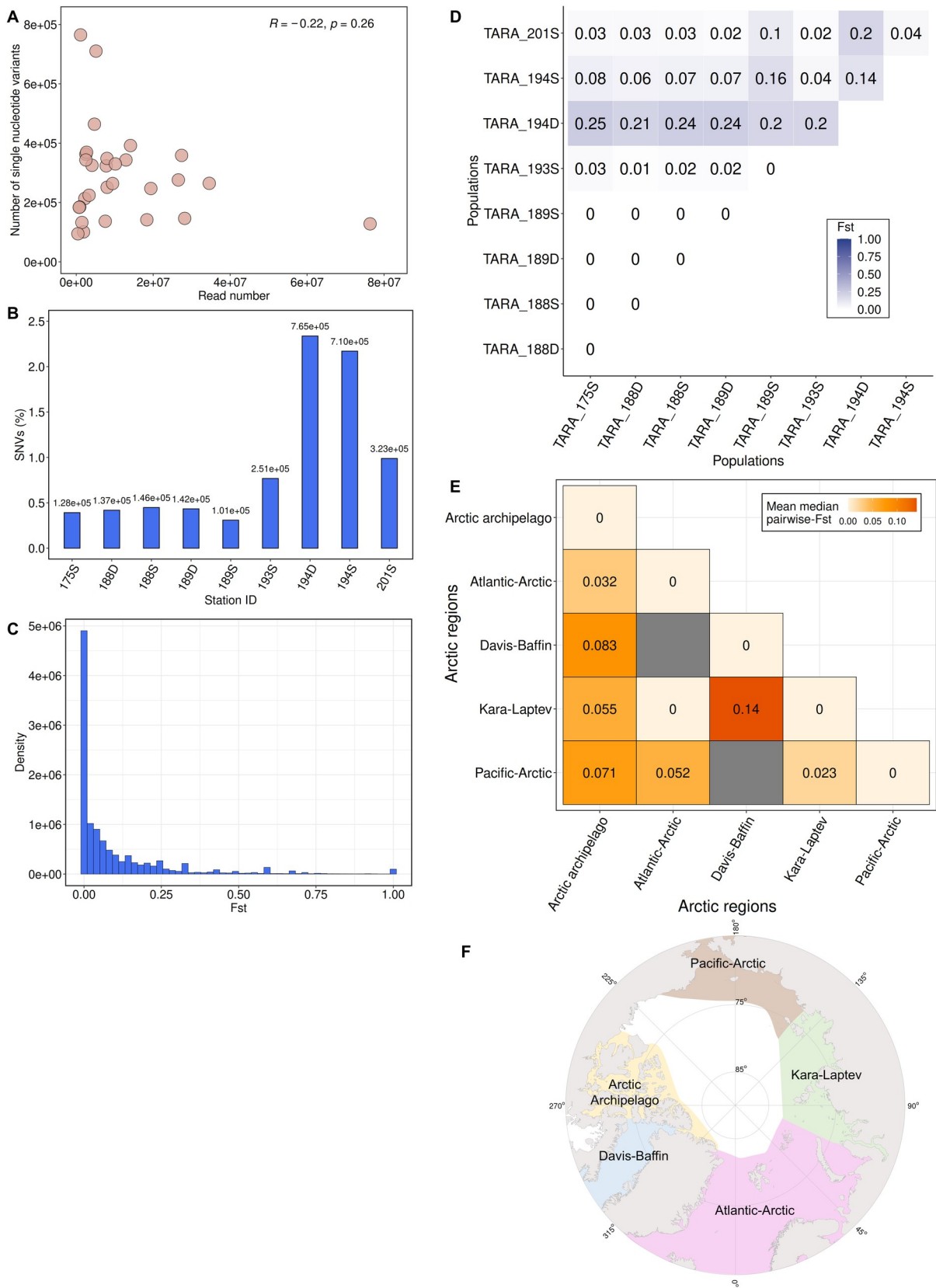

**Fig 5. Population genomic analyses of *Chaetoceros* MAGs.** (A) Scatterplot representing the number of SNVs compared to the number of reads for all samples considered in this study with Pearson's correlation *rho* (*n* = 29). (B) Relative number of SNVs within ARC_116 populations. (C) $F_{ST}$ distribution profile of ARC_116. (D) Pairwise $F_{ST}$ matrix of ARC_116 populations. (E) Global pairwise $F_{ST}$ matrix of all MAGs among Arctic Ocean regions (refers to ARC_116, ARC_189, ARC_217, PSW_256, and SOC_37). D: deep-chlorophyll maximum; S: surface. (F) Polar view of the Arctic Ocean regions (based on [48]). The basemap was generated from the Natural Earth data (http://www.naturalearthdata.com/downloads/50m-physical-vectors/); Natural Earth—All maps are public domain (http://www.naturalearthdata.com/about/terms-of-use/). MAG, metagenome-assembled genome; SNV, single-nucleotide variant.

did not observe a significant correlation between the amount of SNVs and latitude but we noticed a rather strong correlation between SNVs and longitude (S10 Fig and Table A in S5 Table), which might be explained by the effect of oceanic currents in the Arctic Ocean. The most elevated mean SNV level depending on the MAG oceanic regions was observed for the *Chaetoceros* populations in the Pacific South Eastern Ocean (1.21%), followed by those in the Arctic Ocean (1.16%), the Southern Ocean (0.81%), and the Mediterranean Sea (0.76%). Transition to transversion ratios ranged from 1.31 (ARC_217) to 1.96 (PSW_256), with a global average of 1.50 (S11 Fig and Table B in S5 Table). Overall, most of the *Chaetoceros* population variants were observed in the coding regions (49.28% mean value), followed by the intergenic (32.54%), UTR (14.11%), and intronic (4.07%) regions, a pattern consistently observed independently of the genome considered (S12 Fig and Table C in S5 Table). The variant effects were mostly missense mutations (53.29% mean value), followed by silent ones (45.97%), and a slight proportion of nonsense mutations (0.74%) (S13 Fig and Table D in S5 Table).

**Analysis of *Chaetoceros* population structure.** We then explored the level of population structure of the MAGs using the previously identified SNVs associated with the different populations and computed their pairwise fixation index (F-statistic or $F_{ST}$). This index, which can range from 0 (no genetic differentiation) to 1 (complete differentiation), measures the extent of genetic inbreeding between populations using allele frequency, and is thus a proxy of their genetic distance [49,50]. Among the detected variable loci, we selected the SNVs associated with the different populations for the MAGs that were present in at least 2 different sampling points (*Tara* Oceans stations and/or depths), i.e., for ARC_116 (9 samples), ARC_217 (4), ARC_232 (2), PSW_256 (3), and SOC_37 (5). Plotting the population-wide $F_{ST}$ distributions revealed globally unimodal patterns, indicative of a single species for each MAG (Figs 5C and S14 and Sheets A–E available in https://doi.org/10.5281/zenodo.7189810). ARC_116, which was the largest extant MAG, showed populations from stations TARA_175, TARA_188, and TARA_189 appearing to be genetically similar (pairwise $F_{ST}$ of 0), indicating that they formed 1 homogenous population, which is consistent with their respective short distance and closeness to Atlantic water inflow (S15 Fig), potentially favouring elevated migration (Fig 5D and Sheet F available in https://doi.org/10.5281/zenodo.7189810). Noticeably, this MAG showed great (≥0.15) genetic differentiation at the DCM of station TARA_194, according to Wright's guidelines for analysing bi-allelic loci [50], and surface population at the same station also displayed pairwise $F_{ST}$ values distinct from the others but of lower magnitude, ranging from moderate to high (approximately 0.05 to 0.15) genetic differentiation. This difference with the other ARC_116 populations may be at least partly explained by local marine currents, as station TARA_194 is influenced by inflow waters from the Pacific Ocean through the Bering Strait, and TARA_193 is enriched in cold waters circulating back to the Pacific Ocean (S16 Fig). As evidenced by a pairwise $F_{ST}$ of 0.14, both TARA_194 depths appeared to have moderate genetic differentiation among one another. This result suggested that this *Tara* station potentially harboured 2 subpopulations. Examining the metadata associated with this station revealed that the DCM was sampled 30 m deeper (35 m) than the surface (5 m) (S6 Table), with distinct patterns of oxygen concentration and salinity between the depths as well as a

phosphate enrichment at the DCM. One explanation could be that the population of TARA_194 is genetically distinct from the others due to the fact that the Pacific water mass is temporarily interleaved with those more specific to the Arctic. However, in the absence of a temporal reference, it is difficult to conclude. The MAG ARC_217 showed relatively low pairwise $F_{ST}$ values indicating elevated connectivity between the populations, except between stations TARA_205 and TARA_173 (both SUR) located in the Davis-Baffin Bay and in the Kara-Laptev Seas (S15 and S16A Figs and Sheet G available in https://doi.org/10.5281/zenodo.7189810). Both genomes ARC_232 and PSW_256 showed genetically similar populations (S16B and S16C Fig and Sheets H and I available in https://doi.org/10.5281/zenodo.7189810), with PSW_256 exhibiting populations in stations TARA_113, TARA_119, and TARA_120 equally distinct from one another genetically. This pattern might be linked to these stations being located between the Gambier Island archipelago in French Polynesia that may favour high local gene flow. Finally, the SOC_37 genome displayed globally low genetic differentiation (Sheet J available in https://doi.org/10.5281/zenodo.7189810), albeit slightly more elevated when compared with station TARA_173 at the surface than with the others. No clear differentiation with station TARA_201 was observed, although it is located at the opposite side on the Davis-Baffin Bay (S15 and S16D Figs), suggesting a low effect of dispersal on their connectivity.

**Global population structure among Arctic Ocean regions.** We further compared the genomic differentiation of *Chaetoceros* populations between the Arctic regions, which were divided into 5 groups depending on their localisation based on [51]: Pacific-Arctic, Kara-Laptev, Atlantic-Arctic, Arctic Archipelago, and Davis-Baffin. The most elevated genomic differentiation was between the Kara-Laptev and Davis-Baffin regions, which consistently appear opposite from one another (Figs 5E and 5F and S15 and Sheet K available in https://doi.org/10.5281/zenodo.7189810), suggesting lower gene flow compared to the other regions. The *Chaetoceros* populations located in the Arctic Archipelago displayed globally moderate genetic differentiation, with the largest difference being with the Davis-Baffin population. Low genetic structure was observed compared to the populations from the Atlantic-Arctic and Pacific-Arctic, both of which are located in zones with water influx from either the Atlantic or Pacific Oceans (Fig 5F). Low differentiation was also noted between the Kara-Laptev and Pacific-Arctic. Finally, no genetic differentiation was observed between populations in the Kara-Laptev and the Atlantic-Arctic regions, indicative of elevated gene flow between the 2 regions. This was rather expected given that a unique *Tara* Oceans Station in the Atlantic-Arctic, TARA_175, appeared to harbour *Chaetoceros* populations in the present analysis and is located at the interface between the Kara-Laptev regions (Figs 5F and S15).

## Examining the correlation between abiotic parameters and population structure

The above results show that, depending on the MAG considered, there are noticeable patterns of population structure among *Chaetoceros* populations. We then investigated the correlation of different environmental parameters and geographic distance with the genetic differentiation of the MAG populations. For this, we selected the MAGs that were present in at least 3 different stations or depths and with a genetic variance superior to zero, namely, MAGs ARC_116, ARC_217, and SOC_37, all of them having populations in the Arctic Ocean. Pairwise-$F_{ST}$ values between the MAG populations were modelled depending on a range of environmental parameters and Euclidean distance by applying a linear mixed model (LMM), as described in [52], to perform a variance partitioning analysis (Fig 6). We further applied Mantel tests independently on each environmental parameter to verify these results (Sheets A–F in S7 Data).

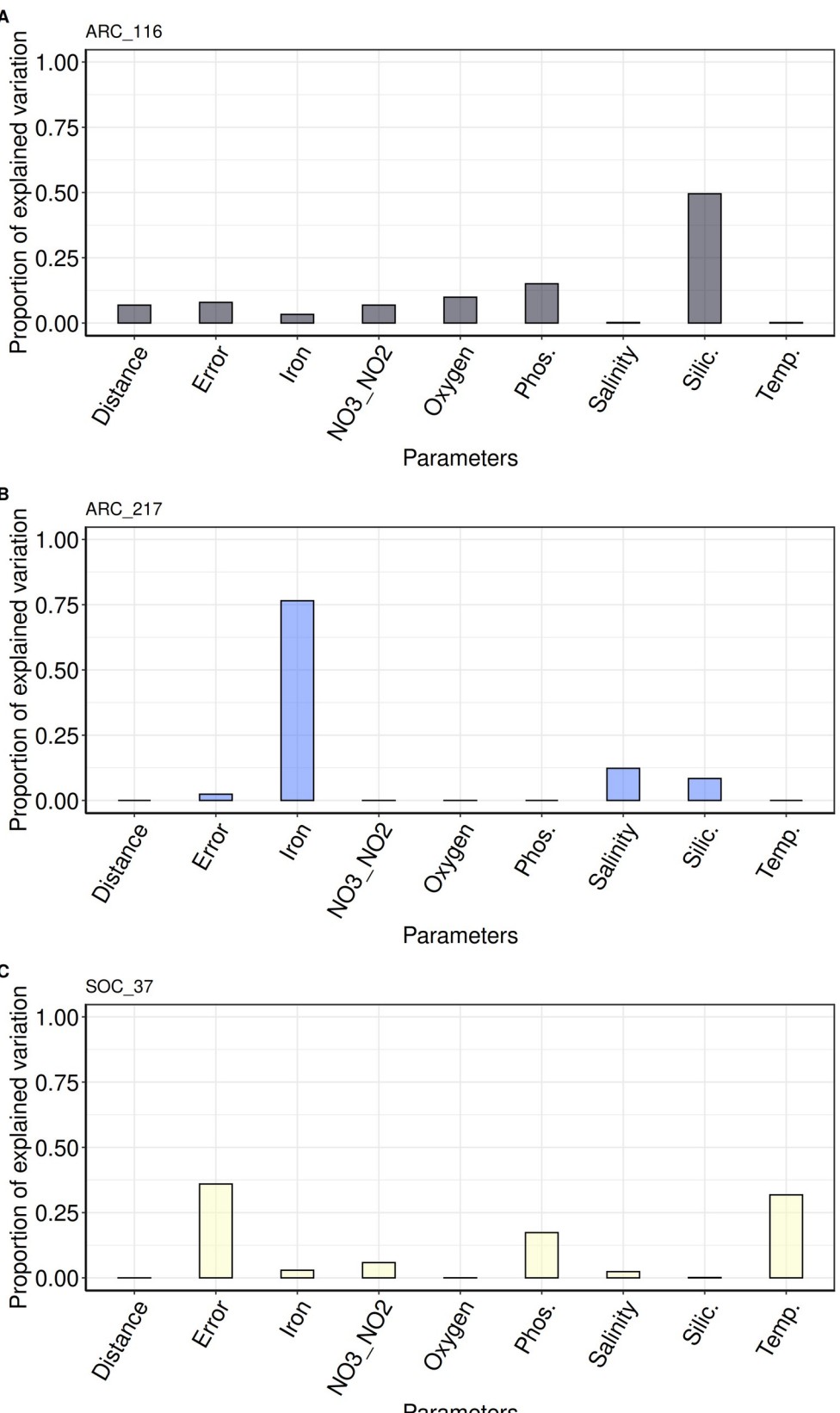

**Fig 6. Correlation between environmental parameters variation and *Chaetoceros* genomic differentiation.** Bar plots of variance partitioning analysis results for (A) ARC_116, (B) ARC_217, and (C) SOC_37 (see Sheets G–I in S7 Data for details). NO3_NO2: sum of nitrate and nitrite concentrations; Phos.: phosphate; Silic.: silicate; Temp.: temperature.

The fixed part of the unexplained variance was below 10% for the analyses involving ARC_116 and ARC_217, and was therefore considered negligible. For ARC_116, most of the genomic variation was correlated with silicate concentrations (50%), followed by phosphate (15%), but they were not validated by the Mantel tests (Figs 6A and S17A and S17B and Sheets B, C, and G in S7 Data). A small correlation with the geographic distance was noted (7%), which was validated by the Mantel test (S17C Fig and Sheets A and G in S7 Data).

On the other hand, iron was the environmental parameter correlated the most (77%) with genomic differentiation of ARC_217, which was not significantly validated by a Mantel test (Figs 6B and S18 and Sheets D and H in S7 Data). Finally, phosphate (45%) and temperature (34%) were the most correlated with genomic differentiation of SOC_37 populations (Fig 6C and Sheet I in S7 Data). We must, however, point out that the model error reached 36%, indicating that the model does not apply well to the populations of this particular MAG and that the predictions are to be taken with caution. Phosphate was not validated by the Mantel tests but temperature was (S19A and S19B Fig and Sheets E and F in S7 Data). The fact that some of the Mantel tests were not validated despite a strong correlation of 1 parameter in the variance partitioning analyses was expected given that most of our samples were small, particularly for ARC_217 and SOC_37. It is indeed evident that most of the data points for these 2 MAGs are fairly dispersed around the regression curve, with minor exceptions (S17–S19 Figs). Moreover, Mantel tests may sometimes give biased *p*-values given the autocorrelation of some environmental variables examined in ecology studies [53,54]. Indeed, a closer investigation of the environmental variable colinearity degree showed that despite our normalisation efforts some of them were correlated (see Materials and methods). Taking this into account, some Mantel tests nonetheless confirmed the correlation between abiotic parameters and genetic differentiation of *Chaetoceros* populations observed in the variance partitioning analyses. From this, we conclude that micro-diversification seems to correlate with different environmental factors in at least some closely related *Chaetoceros* populations.

## Identification of *Chaetoceros* genes under selection

Given the correlation observed between abiotic parameters and the *Chaetoceros* population structure patterns, we then examined whether some genes were undergoing selection. This analysis was conducted on the MAGs presenting variants in at least 3 different populations with discriminant $F_{ST}$ values, i.e., MAGs ARC_116, ARC_217, and SOC_37, using the B-allele frequency (BAF) of the genome variants. The LK distribution of the respective loci followed the expected chi-square distribution (S20 Fig and Sheets A–C available in https://doi.org/10.5281/zenodo.7189835), indicating that the loci followed the neutral evolution model of a single species. We identified several strong candidates potentially under positive selection on the genome contigs, that is 28 loci for ARC_116, 1,116 loci for ARC_217, and 1,101 loci for SOC_37, representing 0.17% (28); 6.89% (802) and 6.60% (534) of the genome contigs, respectively (Tables A–C in S7 Table). Globally, an inspection of the BAF showed that loci under selection were derived principally from *Tara* Oceans stations 175, 188, 193, and 201 (all SUR) for ARC_116, while it was mainly *Tara* Oceans station 201 (SUR or DCM) for ARC_217 and stations 173 and 201 (both DCM) for SOC_37.

We examined the Gene Ontology (GO) terms generated during the Interproscan analysis to gain insights into the functional repertoire of the genes under selection (Sheet A in S8 Data). The ARC_116 MAG displayed a more elevated proportion of genes associated with membrane domains (S21 Fig). Regarding molecular functions, the 3 MAGs displayed an elevated proportion of genes associated with binding and catalytic activities, and ARC_217 showed the most diverse GO terms in this category. The GOs associated with biological processes were mostly represented by general cellular and metabolic processes for all three genomes, followed by cellular organisation or biogenesis as well as localization.

To focus our analysis on describing the potential functions associated with the genes under selection, we selected the SNVs located within a gene sequence with an assigned PFAM domain. A total of 31 PFAM domains were found in the genes harbouring the loci under selection for ARC_116, while we identified 697 for ARC_217 and 805 for SOC_37 (Tables D and E in S7 Table). Among the PFAMs associated with ARC_116 loci 54.84% (17) presented an associated GO term while the percentage was lower for ARC_217 (46.92%) and SOC_37 (40.75%). Most of the outlier loci were associated with coding or untranslated regions (UTRs), with a minor contribution of loci within intronic regions (S22 Fig and Sheet B in S8 Data). Some variants were responsible for loss of function events, such as stop codon gain and frame-shift mutations. We subsequently searched for domain functions potentially linked to the environmental parameters possibly driving the micro-diversification patterns.

Strikingly, all variants under selection within the ARC_116 populations were completely absent from station TARA_189 (DCM). Most of the non-synonymous loci displayed domains associated with kinases, oxidoreductases, and transferases. Among the loci under selection were domains involved in redox balance, such as missense or 3′ UTR variants in genes harbouring glutaredoxin (PF00462, PF13417) and cytochrome *c* oxidase domains (PF02683), all mostly fixed in surface populations of stations TARA_175, TARA_193, and TARA_201 (Table D in S7 Table). Other loci included synonymous, missense, and 3′ UTR variants in domains involved in intracellular transport (e.g., PF04811 and PF08318). Both these domains appeared associated with endoplasmic reticulum to Golgi transport and their variants were almost fixed at the surface of stations TARA_175, TARA_188, TARA_193, and TARA_201.

Conversely, both ARC_217 and SOC_37 showed loci under selection for potential chlorophyll-binding and CobW proteins. The former are found in light-harvesting complexes of the photosynthetic apparatus while the latter form a large family of metal chaperones associated with metal homeostasis processes either with zinc, iron, or cobalt molecules [55,56]. Interestingly, both these functions may have a link with iron availability status, as phytoplankton can cope with iron limitation through remodelling of light-harvesting complexes, while some CobW proteins may exert iron-responsive patterns [57,58]. Comparing the frequency of these variants among the 2 genomes, both ARC_217 and SOC_37 populations were found at stations TARA_201 (Arctic Archipelago) and TARA_173 (Kara-Laptev) and showed elevated frequency of this variant at the former, whereas those at station TARA_173 exhibited lower frequency (Tables E and F in S7 Table). Looking at the environmental metadata of these stations using the PANGAEA database [59–61], we observed that station TARA_173 was characterised by higher iron and nitrate levels but was lower in phosphate (S6 Table). ARC_217 populations were also found at station TARA_205 (SUR), which displayed the lowest iron concentration (51% and 38% lower than the ones of stations TARA_173 and TARA_201 (both SUR), respectively, S6 Table), where the CobW variant was completely fixed and the chlorophyll-binding domain was completely absent. Moreover, we identified a synonymous variant of ARC_217 located in gene TARA_ARC_108_MAG_00217_000000002161.2.2 within a flavodoxin domain (PF00258). Flavodoxin proteins are known to over-accumulate in iron-limited conditions over their iron-containing counterpart ferredoxin [62]. This SNV was almost fixed at the

surface of stations TARA_201 and TARA_205 (Table E in S7 Table). A variant within an iron–sulfur cluster-associated domain (PF02657) involved in redox and regulation of gene expression processes [63] was also found almost fixed in TARA_173 (DCM) and TARA_201 (SUR) SOC_37 populations (Table F in S7 Table).

Most notably, 1 variant of ARC_217 appeared to be located in a gene encoding a low iron-inducible periplasmic domain (PF07692). This SNV corresponded to a synonymous mutation located in the coding region of gene TARA_ARC_108_MAG_00217_000000001530.105.1. An examination of the Manhattan plot of this SNV revealed few drafted variants around the loci under selection, indicative of a potential soft selective sweep, and the BAF showed that selection of this variant was occurring on *Chaetoceros* populations from stations with higher iron concentrations (TARA_173 and TARA_201; both SUR), while it was completely absent from TARA_205 (SUR) (Fig 7A and 7B and 7E and S6 Table).

Next, we searched for potential homologue candidates in *P. tricornutum* by aligning the corresponding ARC_217 protein on the Phatr3 proteome [23] using BLASTp. We identified protein B7FYL2 (https://bioinformatics.psb.ugent.be/plaza/versions/plaza_diatoms_01/genes/view/ptri151180) that was previously identified as an iron starvation-induced protein (ISIP) ISIP2a [17] and showed an e-value of $1 \times 10^{-13}$. The corresponding gene located within a genomic region marked by histone posttranslational modification (PTM) in H3K4me2, a mark suggested to associate with expressed genes in *P. tricornutum* [64]. The mark was reduced in nitrate-limited conditions compared to repletion. We further compared homologue sequences among the 8 *Chaetoceros* transcriptomes from Marine Microbial Eukaryote Transcriptome Sequencing Project (MMETSP) that were used for the phylogeny reconstruction (see Results) and retrieved 16 candidate sequences from 7 transcriptomes (see S9 Data). Aligning these sequences to the reference gene allowed us to identify significant identity at the SNV position for 7 (44%) sequences, including 2 sequences from *C. dichaeta* that displayed the same SNV (C > T) as the loci under selection (S23 Fig and S10 Data). Although the SNV observed in the ARC_217 gene sequence was predicted to induce a synonymous mutation, we searched whether the nucleotide substitution modified the predicted RNA secondary structure through an analysis conducted on the LinearFold and RNAfold web servers [65,66], as silent mutations may impact haplotypes as for instance changes in RNA secondary structure [67]. No clear change of RNA secondary structure was predicted by LinearFold (S24A and S24B Fig). RNAfold outputs showed that the reference sequence displayed an ensemble diversity (i.e., an average base-pair distance between all the structures in the thermodynamic ensemble) of 1,514.62, while the mutated sequence showed a value of 1,486.24. Slight differences of free energy minimisation and centroid (structure with the minimal total base-pair distance to all structures in the ensemble) structures (S24C and S24D Fig) were observed. These results suggested a possible minor impact of the mutation on the RNA folding structure.

Next, an examination of SOC_37 loci under selection revealed genes encoding functions associated with phosphate metabolism, such as a putative cytosolic domain of 10 TM phosphate transporter (PF14703), protein and histidine phosphatase domains (PF00481 and PF00300) (Table F in S7 Table). In accordance with its enrichment in RNA polymerase Rpb1 C-terminal domain (see Results), many genes under selection harboured PFAM domains associated with RNA, as for instance an elongation factor of RNA pol II, RNA recognition motifs, and binding domains, in addition to a reverse transcriptase involved in transposable element activity, most of them almost fixed in station 201 (DCM) while absent from station TARA_173 (SUR) and under selection at the DCM of stations TARA_173, TARA_180, and TARA_194 (Table F in S7 Table). Other functions notably included domains potentially involved in methyl transfers and epigenetic mechanisms regulating gene expression (e.g., PF00850, PF00856, and PF08123), all encoded by genes under selection in TARA_173 and TARA_201

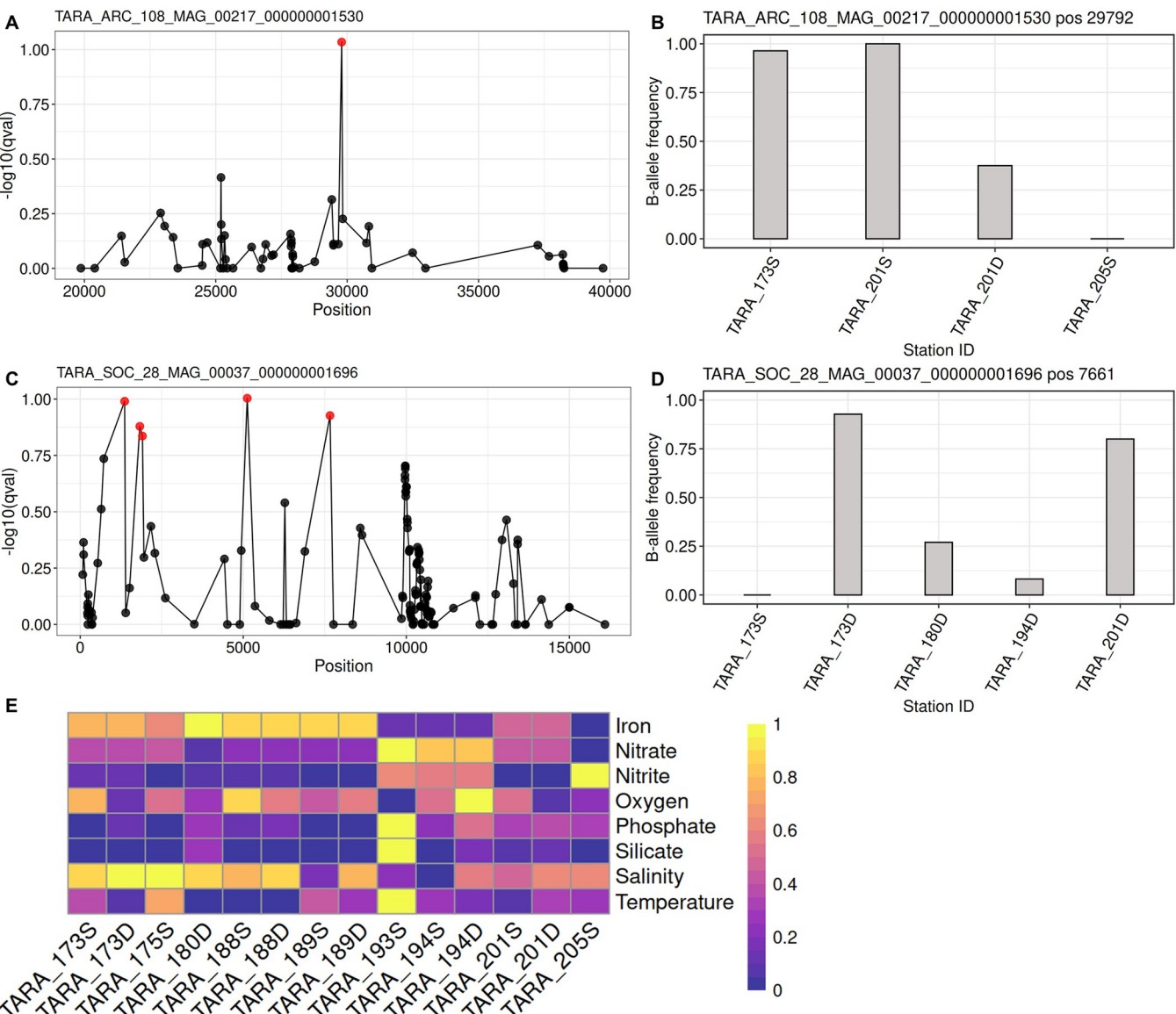

**Fig 7. Selection of variants in Arctic *Chaetoceros* populations.** (A) and (C) represent Manhattan plots in a 10 kb window around the variants of interest, shown for ARC_217 (ISIP) and SOC_37 (spermidine/spermine synthase). Red dots correspond to SNVs considered under selection (*q*-value < 0.15). (B) and (D) represent the bar plots of the BAF for the respective loci of interest depending on the population considered. (E) Range-transformed heatmap of abiotic parameters for the *Tara* Oceans stations in the Arctic Ocean where the *Chaetoceros* MAGs are present (see S6 Table for raw values). The brightest yellow colour represents the most elevated values for any given parameter in the dataset, while the darkest purple indicates the lowest. D: deep-chlorophyll maximum; S: surface. BAF, B-allele frequency; ISIP, iron starvation-induced protein; MAG, metagenome-assembled genome; SNV, single-nucleotide variant.

(DCM). Most remarkably, loci under selection included 2 variants potentially involved in polyamine biosynthetic processes. One SNV was located in the UTR of carbamoyl phosphatase domains (PF00988 and PF02786) and the other in the coding region of a spermine/spermidine synthase domain (PF01564) in gene TARA_SOC_28_MAG_00037_000000001696.30.1, causing a non-synonymous mutation (p.Met1063Leu). Polyamines such as spermine and spermidine are involved in frustule formation through the production of long-chain polyamines [68] using carbamoyl phosphate synthase [16,69]. Inspection of the Manhattan plot of the polyamine synthase variant suggested a potential hard selective sweep signature, and the BAF

showed that it was almost fixed at the DCM of stations TARA_173 and TARA_201 (Fig 7C and 7D) which were characterised by the lowest oxygen concentrations in our analysis (Fig 7E and S6 Table). The carbamoyl phosphate variant showed similar patterns and appeared more frequent in stations with less nitrate and phosphate (S6 Table and Table F in S7 Table). A sequence similarity search for a polyamine synthase homologue in the Phatr3 proteome allowed us to identify the protein B7FPJ4 (https://bioinformatics.psb.ugent.be/plaza/versions/plaza_diatoms_01/genes/view/ptri221670) involved in spermidine biosynthesis [17], with an e-value of $3 \times 10^{-163}$. The gene coding this protein exhibited significant changes of expression in nitrate depletion compared to repletion (-~0.5 fold change; [70]) and in phosphate depletion compared to repletion (approximately 2.9 fold change; [71]). Moreover, this gene located in region marked by H3K9me2 and H3K4me2 PTMs in *P. tricornutum* [64]. Searching among the 8 *Chaetoceros* transcriptomes from MMETSP yielded 17 candidate sequences from 3 transcriptomes (see S11 Data). Among them, 16 (approximately 94%) sequences showed significant identity at the SNV position with 1 sequence from *C. debilis* displaying the same SNV (A > C) as the loci under selection (S25 Fig and S12 Data). Predictions using LinearFold confirmed clear changes of RNA secondary structure between reference and mutated gene (S26A and S26B Fig). Moreover, RNAfold predictions agreed with this pattern as they indicated an ensemble diversity of 957.66 and 1,039.87 for the reference and mutated sequences, respectively, along with significant modifications of free energy minimisation and centroid structure (S26C and S26D Fig). Overall, these findings illustrate the process by which a single mutation may have a direct effect on the electrochemical properties of RNA structures, as well as potentially impacting the biochemical kinetics of the protein.

## Discussion

Our understanding of both the ecology and biological functions of marine algae has progressed considerably with the help of molecular-based methods, generating an increasing number of genomes now reaching more than several hundred [72,73]. Nonetheless, one drawback of laboratory-generated genomes is their potential lack of representativeness of species that thrive in the environment. As an example, it has been demonstrated that *Chlamydomonas reinhardtii* and *P. tricornutum* cultures artificially selected individuals in the population, thus reducing the overall diversity of their genetic pool leading to genetic convergence of the strains; with nutrient-replete conditions favouring somatic mutations leading to the loss of function of some genes that are potentially important in highly fluctuating environments [74,75]. Moreover, interrogating *Tara* Oceans metagenomes with the *Fragilariopsis cylindrus* CCMP 1102 genome showed enough read coverage for only 1 sampling station, exemplifying the divergence between laboratory strains and natural populations [24]. Collectively, these factors may constitute a limit to the understanding of metabolism and locally relevant genomic functions. While cultivating strains in the lab is a necessary first step to gaining insights into their fundamental ecology, accessing the genomes of organisms directly from their environment is key to fully understanding their role within natural communities and their responses to environmental fluctuations. In this context, genome-resolved metagenomics, which consists in the mapping of environmental metagenomic reads on a reference genome, represents a powerful tool to access the diversity and distribution of organisms in their native environment without relying on taxonomic markers that may be too conservative to unveil the amount of diversity, a fact that appears in particular to be the case for unicellular organisms [76]. However, as this field is in its infancy, specific attention must be paid during the binning of metagenomes to prevent chimeric assembly [77]. MAGs must furthermore include information about assembly quality, level of contamination and completeness, to enable robust comparisons between

studies [47]. The exploration of marine phytoplankton population genomics using metagenomes has only just begun, with a few pioneering studies focused on the Mamiellales genera *Bathycoccus*, *Micromonas*, and *Ostreococcus* [78,79]. In line with these, the present study aimed to generate a portrait of the diversity landscape among natural *Chaetoceros* populations and to bridge the gap between diatom genomes, physiological responses, and population genomics by teasing apart the correlation of geographical distance and environmental factors with population structure. Going from not only 1 but 11 metagenomes to gene selection using genome-resolved metagenomics, this work is to our knowledge the first to assess population-scale diversity among *Chaetoceros* genomes directly reconstructed from the environment.

## *Chaetoceros* metagenome-assembled genomes from *Tara* Oceans

The *Chaetoceros* MAGs studied here were generated from *Tara* Oceans metagenomic co-assemblies [46] based on geographically separated samples, which proved better at increasing the global coverage of large genomes but is limited when covering taxa with low local abundance and/or high microdiversity levels. The *Chaetoceros* genus was notably the best represented of all diatom genera (11 out of 52 diatom MAGs) [46], in agreement with its broad distribution and abundance patterns [13]. Future studies involving increased sequencing effort coupled with innovative assembly strategies, such as automated MAG recovery workflows [80], will likely improve the access to new resources for this genus and other diatoms in the near future. The genomes we considered for the present study displayed the same magnitude in size, number of protein-coding genes and G+C content as the newly published *Chaetoceros tenuissimus* genome [81], pointing out the accuracy of the MAG reconstruction methods. Several studies have investigated the link between genome size compared to cell morphology and metabolism and have found that both cell size and growth rate are, respectively, proportional and inversely proportional to genome size [82–87]. We observed contrasted differences in genome sizes for the MAGs ARC_232, ARC_267, MED_399, and PSW_256, which all exhibited the same level of G+C% (the lowest being around 39%), displayed an enrichment of the level of D, E, K, and N amino acids and a depletion in C, D, and A residues, and seemed to belong to the *C. affinis* subclade, indicating that they potentially belong to the same species. These genome size discrepancies suggest potential contrasted growth rates and cell sizes for closely related *Chaetoceros* species. Koester and colleagues [88] noted a 2-fold genome size difference between cryptic but geographically separated populations of the diatom *Ditylum brightwellii*, accompanied by a difference in growth rate, suggesting that whole-genome duplication events may constitute important drivers of genetic diversification in diatoms. Here, both ARC_232 and ARC_267 populations were identified in the Arctic but did not seem to co-occur in our analyses, while MED_399 and PSW_256 were found to be restricted to the Mediterranean Sea and Pacific Ocean, respectively. It is possible that duplication and/or transposition events potentially linked to stress, as was observed in *P. tricornutum* [22], may have given rise to diverged subpopulations that dispersed and, ultimately, were genetically separated following allopatric speciation.

## Insights into biogeographical patterns of the genus *Chaetoceros*

We identified each of the MAGs in a relatively small number of samples with an uneven distribution, suggesting potential habitat specialists, with the notable exception of ARC_116, the only MAG that was distributed in samples spanning globally across the Arctic Ocean, indicating a potential pan-Arctic species. Other studies, based on the *Tara* Oceans and Ocean Sampling Day sample datasets, have already provided a thorough pattern of *Chaetoceros* distribution at global scale in the oceans [13,34]. These have shown a prevalence of *Chaetoceros*

in the Arctic Ocean, with discrepancies depending on the species considered. For example, metabarcoding analyses showed that *Chaetoceros neogracile* was restricted to the northern hemisphere [34], which consistently matches the distribution of ARC_116 and its taxonomic closeness to *C. neogracile* RCC1993. *C. dichaeta* has been retrieved near Alaska and the Antarctic Peninsulahttps://www.zotero.org/google-docs/?b9i6Xk, a distribution that appears in line with that of ARC_189 and with the geographic closeness of PSE_253 in South America. The same study indicated that *C. affinis* was present in the Mediterranean Sea as well as in the Atlantic Ocean and North Sea. We found 4 MAGs (ARC_232, ARC_267, MED_399, and PSW_256) that exhibited taxonomic closeness to *C. affinis*. Only one of them was found in the Mediterranean Sea while the 3 others were retrieved from the Pacific and Arctic Oceans, suggesting potential new niches for this species. *C. debilis*, of which our MAG SOC_60 was also found to be relatively close, was retrieved in different localities: in European coastal waters and in the Arctic Ocean for the northern hemisphere, as well as in the Indian and Southern Oceans, in agreement with the distribution of the MAG. *Chaetoceros* has been viewed as a local opportunistic genus [89,90] but the various species we describe here appeared to evolve in sympatry with up to 3 MAG populations in the same sampling stations. A temporal survey of these sampling points could help reveal whether the populations from different species are sympatric on a regular manner or if some competition mechanisms or niche exclusions are observable. Of note is the observation that none of the MAGs belonging to the same subclade were found at the same locations, with the exception of ARC_217 and SOC_37, indicating that geographic closeness is a poor predictor of the genetic relationship between the MAGs. In general, we found most of the *Chaetoceros* populations in the Arctic Ocean, which may be due to their lower sequence coverage in tropical and subtropical waters, as these species are likely to be among dominant phytoplankton in the Arctic [35]. It is moreover evident that other localisations harbour *Chaetoceros* populations, such as for instance in the Southern Ocean where diatoms dominate photosynthetic protist assemblages [13,35]. In the same vein, associations between *Chaetoceros* and tintinnid ciliates have been observed in the Pacific Ocean and Caribbean Sea [42,91]https://www.zotero.org/google-docs/?lxqQIn, which were only partially sampled during the *Tara* Oceans expeditions [13]. Future oceanographic campaigns should help reveal the distribution of *Chaetoceros* populations and the extent of their genomic variability.

## Genetic differentiation among closely related *Chaetoceros* populations is correlated with different environmental variables

We investigated global patterns of population structure and genetic differentiation in *Chaetoceros*, a cosmopolitan diatom found in every major oceanic province, and one of the most diverse. By leveraging metagenomes reconstructed from *Tara* Oceans samples, we drew a comprehensive landscape of the genetic diversity among different populations of this genus and were able to address their level of gene flow through an analysis of their population structure. The levels of micro-diversity observed here, ranging from 0.63% to a maximum of 2.34%, are in line with previous analyses conducted on natural populations of the diatom *Fragilariopsis cylindrus* from *Tara* Oceans station 86, which displayed approximately 2% SNV density [24]. The observation of elevated genetic differences among populations from the same species means that despite their high potential dispersal, *Chaetoceros* diatoms can express significant levels of divergence. Moreover, our variance partitioning analyses revealed that the genetic differentiation between *Chaetoceros* populations seemed to correlate with a combination of different abiotic factors, with only a minor correlation with geographic distance. This is in agreement with previous studies analysing population diversity using microsatellite markers, such as Härnström and colleagues [92] on *Skeletonema marinoi* and Whittaker and Rynearson

[93] on *Thalassiosira rotula*, and contradicts the results found by Casteleyn and colleagues [94] for *Pseudo-nitzschia pungens*. It must be noted that the former 2 are homothallic centric diatoms while *P. pungens* is a heterothallic pennate diatom. Therefore, the historical assumption that geographic distance is the parameter conditioning most microbial genetic diversity appears conflicting in diatoms.

Among the most notable nutrients that regulate diatom populations are nitrate [95], iron [96,97], phosphate [98,99], silicon [100,101], and cobalamin (i.e., vitamin $B_{12}$) [102,103], although environmental controls of diatom populations vary locally due to their cosmopolitan nature. To our knowledge, the closest study to the present one is that of [93], where the authors investigated the correlation of abiotic parameters and geographic distance with *T. rotula* population structure and revealed a correlation with temperature. Here, although the LMM we applied allowed good predictions for 2 out of the 3 MAGs investigated, preventing us from supporting all the variance partitioning results, we observed a correlation of genetic differentiation with phosphate, silicate, and iron concentrations in *Chaetoceros* species, in addition to a correlation between temperature and genetic differentiation in SOC_37 populations. As mentioned previously, increasing the number of samples investigated by improving MAG assembly and recovery from the environment would likely improve the statistical robustness of the model.

Significant population structure was observed among the *Chaetoceros* MAGs, but with relatively moderate between-region differences in the Arctic, as was observed for zooplankton [52], with $F_{ST}$ levels reaching up to $\geq 0.2$. These high levels of genetic differentiation appear approximately close to those described in different *P. tricornutum* accessions (pairwise $F_{ST}$ approximately 0.2 to 0.4), represented for the most part by strains that have been maintained in culture collections for decades [75]. In particular, ongoing speciation of the ARC_116 population located at station TARA_194, particularly at the DCM, might be a reason explaining why we observed a dramatic number of SNVs, leading us to exclude this station in order to perform a more conservative study when identifying genes under selection. Indeed, this indicates unequal gene flow among the populations and suggests a metapopulation structure consisting of populations of populations, as has been described for the diatom *D. brightwellii* [104]. This observation might stem from a mix of Pacific and Arctic populations coming from intermixed water masses. However, this difference in genetic structure of the ARC_116 population at station TARA_194 was not observed in other populations present at this particular station, such as for SOC_37. Overall, all 3 MAGs ARC_116, ARC_217, and SOC_37 appeared closely related in our phylogenetic analyses but nonetheless showed elevated numbers of MAG-specific orthogroups, and their genetic differentiation was correlated with different sets of abiotic parameters. Taken together, these results emphasise that even with the same local environmental conditions, populations of closely related diatoms from the same genus seem to display different gene flow patterns, emphasising their enormous genetic diversity as well as significant adaptive potential.

### Functional overview of natural selection among *Chaetoceros* populations

We were able to identify genes under selection between the different *Chaetoceros* populations and tried to assess their respective functions. Previous studies have investigated the gene functions that are essential for diatom survival. Among these are functions associated with light perception and energy dissipation, such as for instance phytochromes involved in red/far-red light sensing [105] and light-harvesting complex stress-related proteins (LHCX1) that modulate light responses [106]. Other important functions include metabolic plasticity and response to nutrient fluctuations. As an example, ornithine-urea cycle proteins mediate rapid responses

to nitrogen variability [69]. Another remarkable characteristic of diatoms is their ability to respond to iron fluctuations, as they are among communities most strongly linked to the concentration patterns of this micronutrient [97]. Indeed, diatoms exhibit a diverse range of iron uptake mechanisms, involving siderophores [107], phytotransferrins [108,109], and ferric reductases [110]. They exhibit differential mechanisms of iron storage, with *Pseudo-nitzschia* using ferritin while members of the *Chaetoceros* and *Thalassiosira* genera are believed to be able to store iron in their vacuole [111]. While we did not observe selection of functions related to light acquisition in our analyses, clear positive selection patterns of genes involved in iron responses were retrieved, for the most part in ARC_217 populations. These included a gene encoding an ISIP, which exhibited contrasted frequency that followed iron concentration patterns, the variant being more frequent in stations with more elevated iron concentrations. We found this gene to be a homologue of *P. tricornutum* ISIP2a, which encodes a protein involved in concentrating ferric iron at the cell surface [109], with a function equivalent to human transferrin, hence its name "phytotransferrin" [112]. This protein has been proposed to constitute an ecological marker of iron starvation in diatoms [113] as it is strongly up-regulated by this condition. Previous analyses in *P. tricornutum* cells deficient in ISIP2a showed reduced iron uptake capabilities [107,109]. While our results suggested a slight effect of the mutation on its RNA secondary structure, we identified the same SNV in sequences homologous to this gene in *Chaetoceros* transcriptomes. It therefore appears plausible that relaxed selective pressure on the gene in an environment more iron-replete could have led to the observed allele.

Additionally, we noted the positive selection of genes encoding a carbamoyl phosphate synthase and spermine/spermidine synthase in SOC_37 populations, with the latter showing a potential impact on RNA secondary structure, along with other genes linked with phosphate metabolism. Carbamoyl phosphate synthase is thought to catalyse the first step of the urea cycle, a process that generates polyamine precursors [16]. Polyamines such as spermine and spermidine are nitrogenous compounds involved in frustule formation through their interaction with the heavily phosphorylated silaffin peptides [6,114]. Consequently, diatom frustule formation relies on both nitrogen and phosphorus. We identified a polyamine synthase homologue in *P. tricornutum* displaying significant modulation of its expression in response to nitrate and phosphate availability levels, as well as a homologue bearing the same allele in a *Chaetoceros* transcriptome. Polyamine biosynthetic processes have been linked to diatom physiological responses to nitrogen, salinity, and temperature [115–117] and many polar diatoms show increased silicate content under iron-limiting conditions, which can result from either increased silicate accumulation or lower accumulation of nitrate depending on the species considered [118,119]. Despite this, the putative link between abiotic parameter variations across stations and the variant frequency patterns observed remains unclear. In this study, we noted that several genes harbouring SNVs displayed homologues associated with PTMs in *P. tricornutum* [64]. Changes in the amount of cytosine residues in genes (e.g., A > C) could potentially affect gene expression through the increase or decrease of methylation sites available. Overall, future studies involving engineered knock-out mutants of the genes containing the SNVs and including different sets of abiotic parameters should help gain insights into their respective impacts on gene expression patterns and whole-cell fitness.

## Conclusion

*Chaetoceros* is the most widespread and connected diatom genus, making it a key component of plankton communities, and has been identified as a genus vulnerable to projected climate change. Here, we have observed significant correlations between nutrient availability and genetic differentiation among some *Chaetoceros* populations, with a potential impact on their

gene selection and, in turn, growth strategies. As climate change is expected to influence water stratification, acidification as well as nutrient availability, it appears more than likely that predicted environmental changes in the Arctic will influence *Chaetoceros* distributions and its gene pool. The present study positions itself as an extension of previous work realised on plankton population genomics, and is to our knowledge the first of its kind dealing with micro-diversification patterns of multiple MAGs from a non-model diatom genus. Finally, this work highlights the necessity to perform repeated sampling over time to be able to test whether separated *Chaetoceros* populations can evolve in sympatry as well as the effect of seasonality and local environmental fluctuations on gene selection.

## Materials and methods

### Genomic resources

The *Tara* Oceans metabarcoding V9 18S rDNA data (*n* = 2,739 OTU barcodes) were extracted from [120] (available in https://zenodo.org/record/3768510#.Ye-xqpHMJzp). The 4 eukaryotic-enriched size fractions 0.8 to 5 μm, 5 to 20 μm, 20 to 180 μm, and 180 to 2,000 μm were pooled and the surface (SUR) and DCM depths separated (see S1 Data). The R package "mapdata" v2.3.0 was used as base layer for plotting the maps (http://cran.nexr.com/web/packages/mapdata/index.html). Eleven reconstructed and manually curated MAGs, generated from *Tara* Oceans metagenomic reads [46]; available at https://www.genoscope.cns.fr/tara/#SMAGs and belonging to the genus *Chaetoceros* were considered in the present study. For clarity and readability, the original MAG IDs were shortened and the corresponding information is available in S1 Table. Information corresponding to the genome size and number of genes were extracted from [46]. All these genomes received a former geographical assignment based on their read recruitment in the *Tara* Oceans sampling stations after mapping each MAG onto the *Tara* Oceans metagenomic dataset. The estimation of genome completion was performed by retrieving the Benchmarking Universal Single-Copy Orthologs (BUSCO) [121] using the DNA contigs of the MAGs and control genomes as input for BUSCO v2.0.0 with the eukaryota_odb10 library (*n* = 255 BUSCOs, see S2 Table for complete summary). Gene length was estimated using SAMtools [122] "faidx" on the genome FASTA files. Percentage of G+C in the genome codons was calculated with the COUSIN online tool [123]. Reference diatom genomes for *P. tricornutum* CCAP 1055 and *T. pseudonana* CCMP 1335 were also considered to provide a comparison with the *Chaetoceros* MAGs and were both retrieved from the Joint Genome Institute [16,17]. The genome of *Chatoceros tenuissimus* NIES-3715 was retrieved from the DDBJ repository (https://www.ddbj.nig.ac.jp) [81]. Information about their completion level, gene length, and percentage of G+C were obtained with the aforementioned methods, except for *C. tenuissimus* for which the genes were not available.

### Average nucleotide identity and average amino acid identity

ANI between the MAGs was calculated using FastANI [124] on the whole genomes with the—ql and—rl parameters and—minFraction 0.05 to retrieve all the identity percentages even for highly divergent MAGs. In a similar manner, AAI was estimated on the whole predicted proteomes using the online AAI calculator provided by the Konstantinidis lab (http://enve-omics.ce.gatech.edu/aai/) [125].

### Phylogeny and identification of orthogroups

To investigate the taxonomic relatedness of the MAGs, we considered the BUSCO genes identified from estimation of genome completion (see above). Using the BUSCO IDs for which at

least 8 out of the 11 MAGs (approximately 70%) presented a sequence (83 over 255 eukaryotic BUSCOs), the BUSCO gene clusters translated into proteins were aligned with MAFFT [126] in automatic mode, followed by a manual cleaning of each of the alignments by removing N-terminal and C-terminal residues displaying less than 70% conservation. The alignments were then trimmed using trimAl [127] with the parameter -gt 0.5, followed by another alignment step with MAFFT. A total of 83 gene clusters were retained. To identify potential contaminants, consensus guide trees were generated for each of the approved alignments using RAxML [128] with the parameters -m PROTGAMMAJTT for the substitution model, -N 100 bootstrap replicates and randomly defined numbers between 1 and 99,999 for the parameters -x and -p. To evaluate the MAGs relatedness with respect to other taxa, the same approach was conducted based on a concatenation tree with 23 supplementary taxa sampled across the eukaryotic tree of life (34 total taxa), including 8 *Chaetoceros* transcriptomes from MMETSP deposited in the European Nucleotide Archive converted into protein sequences, for the same 83 single-copy nuclear genes. The cleaned alignments were then concatenated and a final ML tree was built with RAxML 8.2.12 (100 bootstraps) and the final figure exported in iTOL [129]. Two analyses with the OrthoFinder [130] software were conducted to identify the orthogroups: a first one using the protein sequences from the MAGs (available at https://www. genoscope.cns.fr/tara/#SMAGs) and the 23 other taxa, in which 846 orthogroups including at least 50% of species having single-copy genes in any orthogroup were used as input to build a species tree in iTOL, and a second one considering only the *Chaetoceros* MAGs to identify their orthologous genes.

## Comparative analysis of the amino acid composition and PFAMs of the MAGs

The amino acid composition of the genomes was computed by analysing the FASTA files with the "protr" package in R [131] and plotting their corresponding frequencies. The amino acid composition of each MAG was then normalised by the respective amino acid global mean. Protein FAMilies (PFAMs) domains were inferred by searching in a local installation of Interproscan [132]. Raw values are available in S3 Table.

## Genome-resolved metagenomics of *Chaetoceros* MAGs

To generate an estimate of *Chaetoceros* MAG abundance, we performed a mapping of the *Tara* Oceans metagenomic dataset on their contigs using BWA-MEM [133] with default parameters, with an 80% identity filter and at least 4× mean vertical coverage, removed the duplicate reads, and stored the recruited reads as BAM files with SAMtools. The resulting mapping files were sorted using SAMtools "sort" and the different size fractions were merged to increase the coverage, keeping the SUR and DCM depths separated in order to compare their corresponding local populations when possible. Read identity was extracted using a custom perl script using SAMtools "view," and plotted in RStudio. We performed a final step of read filtration on their identity by extracting the read names with at least 97% identity in RStudio, then retrieving them using the Picard toolkit (https://broadinstitute.github.io/picard/) on the indexed BAM files with the option "FilterSamReads" in "lenient" mode. In order to ensure that the recruited reads belonged to our genomes of interest, we generated plots of the genomic coverage of the reads with Bedtools "genomecov" [134] and performed a visual inspection to detect bimodal trends. We excluded the reads from sampling stations that presented insufficient or bimodal coverage distribution (see S6 Fig where we report an example of discarded samples based on read coverage), and the ones with a coverage breadth estimated with SAMtools "depth" inferior to 80. These read filtration and sample inspection steps were critical as

some nonspecific read recruitment may happen (due for instance to the stability of 18S rRNA gene and/or to hypervariable genomic regions). This resulted in a total of 20 *Tara* Oceans Stations and/or depths investigated. The relative abundance of the MAGs in each sample was estimated as the number of mapped reads, obtained with SAMtools "flagstat," normalised by the total number of reads per sample (available in Supporting information Table S4 of [46]). We then computed the relative proportion of *Chaetoceros* reads per station and generated the corresponding maps with RStudio 4.0.1. Pairwise correlation of the MAG relative abundances across the *Tara* Oceans stations was investigated using the "corrplot" R package (https://github.com/taiyun/corrplot).

## Population genomics analyses

Genomic variants of the *Chaetoceros* populations associated to different stations and/or depths were called with BCFtools 1.13.25 [135] to generate a VCF file (mpileup of the files with at least 97% identity). Variant annotation of the VCF files was conducted using SnpEff [136], converting GFF files into GTF 2.2 format with AGAT [137]. The number, position, and type of variants as well as their respective effects were plotted on RStudio.

To calculate the genetic distance between the MAG populations, the genetic variants were identified using SAMtools mpileup -B (multiple BAM files per MAG) and merged into 1 file. The PoPoolation2 tool [138] was then applied on the MAGs present in at least 2 stations (ARC 116, ARC 217, ARC 232, SOC 37) to generate a synchronised (.sync) file from the merged mpileup with the following parameters:—fastq-type sanger—min-qual 20. The.fst files were subsequently computed with the parameters—suppress-noninformative—min-count 2—min-coverage 4—max-coverage 200—min-covered-fraction 1—window-size 1—step-size 1—pool-size 500. The $F_{ST}$ metrics were computed from the allele frequencies (not the allele counts) using the equation in [139]. Allelic frequencies were computed with PoPoolation2 with the parameters—min-count 2—min-coverage 4—max-coverage 200. To ensure that these alleles belonged to our query genomes, the global population-wide F-statistic was computed for each MAG of interest and its distribution plotted and inspected for unimodality. Median pairwise-$F_{ST}$ values were considered as a proxy for genomic differentiation between the respective MAG populations. In addition, the LK statistics [140] were computed and compared with the expected chi-squared distribution with df = *n*-1, with *n* being the number of populations. Under a neutral evolution model, the loci are supposed to follow an expected chi-squared distribution if there is a single species.

We further investigated the global connectivity level between the MAG populations in the different Arctic Ocean regions. For this, the Arctic stations were divided based on their geographic location, as was previously done by [51]https://www.zotero.org/google-docs/?oBViXj. Five regions were identified: Pacific-Arctic, Kara-Laptev, Atlantic-Arctic, Arctic Archipelago, and Davis-Baffin (see Figs 5F and S15 for the polar view of the Arctic *Tara* Oceans stations). To compare the level of genomic differentiation between these regions, median between regions pairwise-$F_{ST}$ values from all the *Chaetoceros* populations were extracted and their respective means compared. The R packages "oce" v1.7–6 [141] and "PlotSvalbard" v0.9.2 (https://github.com/MikkoVihtakari/PlotSvalbard) were used to build the maps centered on the Arctic Ocean.

## Estimation of variance partitioning

In a second step, the estimation of the correlation between abiotic parameters and geographic distance and the genomic differentiation of MAGs was undertaken. For this, an LMM from the R package "MM4LMM" [142] was used as previously applied by [52] on marine plankton. As an input dataset for abiotic parameters, median values of different environmental

parameters were extracted from the PANGAEA database [59–61] and for each sampling site, namely, oxygen, salinity, and temperature, as well as concentrations of ammonium, iron, nitrate, nitrite, phosphate, and silicate. Environmental variables were z-score normalised (Sheet A in S13 Data). Correlation between variables was inspected with the R package "corrplot" that highlighted several correlated parameters (S27A Fig and Sheets B and C in S13 Data). We therefore removed ammonium and added together nitrate and nitrite into 1 variable called "Nitrate_Nitrite" to avoid redundancy. The resulting correlation plot still showed significant correlation patterns between phosphate, silicate, and Nitrate_Nitrite, and between the latter and iron (S27B Fig and Sheets D and E in S13 Data). We further analysed the multicolinearity among variables using the R package "performance" [143], which confirmed elevated ($\geq$10) variance inflation factors between the $F_{ST}$, Nitrate_Nitrite, and iron matrices in our dataset. We however chose to keep these variables as all of them were shown to significantly influence diatom growth in the environment. Euclidean distances were then computed between the stations for all these parameters as well as for the station coordinates as a proxy for geographic distance. Finally, median pairwise-$F_{ST}$ values with the different abiotic parameters and distances were used as input for the LMM, allowing to estimate the relative proportion of the genomic variance explained by each parameter as well as an unexplained proportion. Mantel tests from the "vegan" R package [144] were applied to verify the results independently on each environmental variable.

## Identification of genes under selection

The "pcadapt" R package v4.0.2 [145] was applied to detect selection among populations using the BAF matrix, on "pool-seq" mode with a minimal allele frequency of 0.05 within the populations, as was done in [52]. Based on the PCA results, 2 samples from 1 station (194 SUR and DCM) of the MAG ARC_116 exhibited very distinct variation patterns (S28A Fig and Sheet A available in https://doi.org/10.5281/zenodo.7189835), leading to a very high number of outliers (12,954) and were therefore removed from the analysis (S28B Fig and Sheet B available in https://doi.org/10.5281/zenodo.7189835) to avoid false-positive inflation. We computed $q$-values using the R package "qvalue" with false discovery rate correction [146]. For the 3 MAGs ARC_116, ARC_217, and SOC_37, loci with a $q$-value <0.15 were considered to be under selection. The functions of the genes harbouring loci under selection were investigated with the PFAM domains generated previously. Homologues of the genes of interest were subsequently searched by reverse genetics in *P. tricornutum* (Phatr3) by BLASTp. Manhattan plots of the contigs harbouring these loci were then built for each MAG, as well as bar plots representing their BAF. To investigate the frequency of selected genes displaying loci under selection in other *Chaetoceros* species, searches of homologues among *Chaetoceros* spp. transcriptomes (derived from MMETSP; see the phylogeny section in Materials and methods and S23 and S25 Figs) were performed by aligning target protein sequences on the 8 transcriptomes using tBLASTn. Output sequences were considered as homologue candidates when their scores and e-values were, respectively, at least equal to 200 and $1 \times 10^{-50}$. Alignments were conducted using the online version of MAFFT v7.463 (https://mafft.cbrc.jp/alignment/software/) [147]. The sequences and alignments are available in the S9–S12 Data. Predictions of RNA secondary structures were conducted using RNAfold 2.4.18 (http://rna.tbi.univie.ac.at/) [65] and Linearfold (beta) (http://linearfold.org/) [66] web servers.

## Supporting information

**S1 Table.** (A) Detailed information about the MAG IDs adapted from Delmont and colleagues (2022). (B) Detailed genomic information of the MAGs. (C) Gene length of the

genomes. (D) G+C percentage of the genomes at the different codon positions.
(XLSX)

**S2 Table. Detailed summary of BUSCO analysis used as input for Fig 2B and for the phylogenetic tree in Fig 3B.**
(XLSX)

**S3 Table. Raw values of the 46 PFAM domains displaying the most variable copy number (SD $\geq$ 10) among the MAGs.**
(XLSX)

**S4 Table.** (A) Relative contribution of the MAGs to total read abundance. D: deep-chlorophyll maximum; S: surface. (B) Raw number of MAG reads and relative contribution to total station reads. D: deep-chlorophyll maximum; S: surface.
(XLSX)

**S5 Table.** (A) Number of reads, SNVs, and SNV percentage. D: deep-chlorophyll maximum; S: surface. (B) Transition to transversion ratio of the MAGs. (C) Variant position of the MAGs. (D) Variant effect of the MAGs.
(XLSX)

**S6 Table. Environmental parameters of the *Tara* Oceans stations investigated.** D: deep-chlorophyll maximum; S: surface.
(XLSX)

**S7 Table.** (A) Loci under selection in ARC_116 populations. D: deep-chlorophyll maximum; S: surface. (B) Loci under selection in ARC_217 populations. D: deep-chlorophyll maximum; S: surface. (C) Loci under selection in SOC_37 populations. D: deep-chlorophyll maximum; S: surface. (D) Loci under selection in ARC_116 populations and located within a gene sequence with an associated PFAM domain. D: deep-chlorophyll maximum; S: surface. (E) Loci under selection in ARC_217 populations and located within a gene sequence with an associated PFAM domain. D: deep-chlorophyll maximum; S: surface. (F) Loci under selection in SOC_37 populations and located within a gene sequence with an associated PFAM domain. D: deep-chlorophyll maximum; S: surface.
(XLSX)

**S1 Fig. G+C content of the MAGs.** (A) Mean G+C content of MAGs and reference diatom genomes *Chaetoceros tenuissimus* (C.t.), *P. tricornutum* (P.t.), and *T. pseudonana* (T.p.). (B) Distribution of G+C content along codon positions of the MAGs.
(PNG)

**S2 Fig.** (A) AAI of the MAGs. (B) Correlation analysis between ANI and AAI showing significant positive correlation (Pearson's correlation, the shaded area corresponds to 95% confidence interval).
(PNG)

**S3 Fig. Species relatedness of the *Chaetoceros* MAGs using orthogroups.** The tree was based on 846 orthogroups containing at least 50% of species having single-copy genes in any orthogroup. Support values are indicated on the branches.
(PNG)

**S4 Fig. Comparative analyses of *Chaetoceros* MAGs and their coding potential.** (A) Upset plot representing the top 30 shared orthogroups among the MAGs, with the orthogroups shared by all genomes highlighted in blue. (B) Frequency of the most variable amino acids

compared to their global means across all MAGs. The MAG respective number of genes is indicated for comparison. (C) Heatmap of 46 PFAM domains displaying the most variable copy number (SD $\geq$ 10) among the MAGs (see S2 Table for details).
(PNG)

**S5 Fig. Number of PFAM domains per gene.** Boxplots plotting the distribution of the number of PFAM domains per gene with the blue dot representing the mean.
(PNG)

**S6 Fig. Example of read coverage distributions.** Read coverage distribution of MAG PSE_171 at the surface of stations (A) TARA_173 and (B) TARA_92. The distribution at station TARA_173 does not display enough coverage depth nor a clear unimodal pattern and will be discarded, whereas the pattern of TARA_92 is clearly unimodal and centered around 30×, hence the reads from this station will be kept for further analyses.
(PNG)

**S7 Fig. Identity profiles of the metagenomic reads for the different *Chaetoceros* MAGs after filtration over 4× mean vertical coverage and 80% identity (plots shown starting at 90% identity for readability).** (A) ARC_116, (B) ARC_189, (C) ARC_217, (D) ARC_232, (E) ARC_267, (F) MED_399, (G) PSE_171, (H) PSE_253, (I) PSW_256, (J) SOC_37, and (K) SOC_60. The blue dots represent mean identity.
(PNG)

**S8 Fig. Bubble plots of different nutrient concentrations at the respective sampling points of the MAGs.** For a given MAG, most of the populations are distributed across a rather wide spectrum of iron, silicate, phosphate, and nitrate concentrations.
(PNG)

**S9 Fig. Number of variants in the *Chaetoceros* populations.** SNV percentages are shown, together with the total number of variants.
(PNG)

**S10 Fig. SNVs and geography.** Scatterplots showing the repartition of the SNVs in the populations regarding their (A) latitude and (B) longitude (Pearson's correlation coefficients and *p*-values are shown). No correlation was observed for the latitude while a negative effect of the longitude was observed.
(PNG)

**S11 Fig. Transition to transversion ratios of the *Chaetoceros* populations.**
(PNG)

**S12 Fig. Relative position of the SNVs in the *Chaetoceros* populations.** The exact proportion of each category is annotated.
(PNG)

**S13 Fig. Relative effects of the variants in the *Chaetoceros* populations.**
(PNG)

**S14 Fig. Population-wide $F_{ST}$ profiles of the MAGs.** The distributions are shown for the MAGs presenting at least 2 populations, that is (A) ARC_217, (B) ARC_232, (C) PSW_256, and (D) SOC_37. The $F_{ST}$ distributions appear globally unimodal, confirming that the reads of the respective MAGs were recruited from a single species.
(PNG)

**S15 Fig. Polar view of the *Tara* Oceans stations located in the Arctic Ocean where the *Chaetoceros* populations were identified.** The *Tara* sampling stations belong to Arctic zones as follows: Atlantic-Arctic (175), Kara-Laptev (173, 180, 188, and 189), Pacific-Arctic (193 and 194), Arctic Archipelago (201), and Davis-Baffin (205) (based on Royo-Llonch and colleagues, 2021). The R packages "oce" v1.7–6 [141] and "PlotSvalbard" v0.9.2 (https://github.com/MikkoVihtakari/PlotSvalbard) were used to build the maps centered on the Arctic Ocean.
(PNG)

**S16 Fig. Pairwise $F_{ST}$ matrices of the respective MAG populations.** The matrices correspond to (A) ARC_217, (B) ARC_232, (C) PSW_256, and (D) SOC_37 populations.
(PNG)

**S17 Fig. Mantel tests of ARC_116.** For (A) silicate, (B) phosphate, and (C) geographic distance.
(PNG)

**S18 Fig. Mantel test of ARC_217.** Shown for iron.
(PNG)

**S19 Fig. Mantel tests of SOC_37.** For (A) phosphate and (B) temperature.
(PNG)

**S20 Fig. Population-wide LK profiles of the MAG loci.** The orange line indicates the $\chi 2$ (df = 6; 3 and 4) theoretical distribution. The observed LK distributions followed the expected one.
(PNG)

**S21 Fig. GO terms associated with the loci under selection in *Chaetoceros* MAGs ARC_116, ARC_217, and SOC_37.**
(PNG)

**S22 Fig. Effect of the loci under selection within a PFAM domain, according to their position, for the MAGs ARC_116, ARC_217, and SOC_37.**
(PNG)

**S23 Fig. Alignment of gene TARA_ARC_108_MAG_00217_000000001530.105.1 with homologue sequences from *Chaetoceros* spp. transcriptomes.** Sequence alignment was realised with MAFFT v7 with the gene 1530.105.1 set as reference sequence. The position of the reference nucleotide undergoing selection in some ARC_217 populations is highlighted. The sequences and alignment are in S1 and S2 Data files.
(PNG)

**S24 Fig. RNA secondary structure predictions for gene TARA_ARC_108_-MAG_00217_000000001530.105.1.** Outputs from the LinearFold webserver for (A) the reference gene and (B) the gene with the p.Leu277Leu mutation showing exactly the same structure patterns. Mountain plots and positional entropy outputs from RNAfold representing the minimum free energy structure for the (C) reference gene and (D) gene with the SNV, displaying slightly different patterns.
(PNG)

**S25 Fig. Alignment of gene TARA_SOC_28_MAG_00037_000000001696.30.1 with homologue sequences from *Chaetoceros* spp. transcriptomes.** Sequence alignment was realised with MAFFT v7 with the gene 1696.30.1 set as reference sequence. The position of the reference nucleotide undergoing selection in some SOC_37 populations is highlighted. The

sequences and alignment are in S3 and S4 Data files.
(PNG)

**S26 Fig. RNA secondary structure predictions for gene TARA_SOC_28_-MAG_00037_000000001696.30.1.** Outputs from the LinearFold webserver for (A) the reference gene and (B) the gene with the p.Met1063Leu mutation showing significantly different structure patterns. Mountain plots and positional entropy outputs from RNAfold representing the minimum free energy structure for the (C) reference gene and (D) gene with the SNV, displaying again different patterns.
(PNG)

**S27 Fig. Correlation analysis of the environmental variables.** (A) Correlation matrix before simplification of the number of variables; (B) correlation matrix after variable simplification (values represent Spearman's correlation *rho*, data are shown for significant correlations with *p*-value < 0.01).
(PNG)

**S28 Fig. Principal component analysis of ARC_116 allelic frequencies in the different populations.** (A) The PCA of all the stations shows that populations from TARA_194 at the surface and DCM are pulled away from the others. (B) The PCA after removing the outlier points from station TARA_194 shows more details on the allelic frequency patterns, with less heterogeneous dispersion. Stations TARA_188D, TARA_189S and TARA_189D are clearly apart from the others.
(PNG)

**S1 Text. Comparative analyses regarding orthogroups, amino acid content, and PFAM domains.**
(DOCX)

**S1 Data. Distribution of *Chaetoceros* V9 18S rDNA reads across *Tara* Oceans.** DCM: deep-chlorophyll maximum; SUR: surface.
(CSV)

**S2 Data.** (A) Pairwise ANI of the MAGs. (B) Pairwise AAI of the MAGs.
(XLSX)

**S3 Data.** Phylogenetic tree of *Chaetoceros* MAGs and other eukaryotes using BUSCO genes, in newick format.
(NEWICK)

**S4 Data.** Phylogenetic tree of *Chaetoceros* MAGs and other eukaryotes using orthogroups, in newick format.
(NEWICK)

**S5 Data.** (A) Orthogroup data of the MAGs. (B) Amino acid data of the MAGs. (C) Number of PFAM domains per gene for the MAGs.
(XLSX)

**S6 Data.** (A) Spearman's correlation value of the MAG abundances. (B) *P*-values of the correlation of the MAG abundances. (C) Environmental parameters and abundance of the MAGs.
(XLSX)

**S7 Data.** (A) Euclidean distances of silicate concentrations for ARC_116 Mantel test. (B) Euclidean distances of phosphate concentrations for ARC_116 Mantel test. (C) Euclidean

distances using geographical coordinates for ARC_116 Mantel test. (D) Euclidean distances of iron concentrations for ARC_217 Mantel test. (E) Euclidean distances of phosphate concentrations for SOC_37 Mantel test. (F) Euclidean distances of temperature for SOC_37 Mantel test. (G) Parameters of ARC_116 variance partitioning analysis. (H) Parameters of ARC_217 variance partitioning analysis. (I) Parameters of SOC_37 variance partitioning analysis. (XLSX)

**S8 Data.** (A) GO terms associated with the 3 MAG variants under selection. (B) Position of the variants under selection for the 3 MAGs.
(XLSX)

**S9 Data. Candidate ISIP-encoding genes.**
(TXT)

**S10 Data. Alignment of the candidate ISIP-encoding genes.**
(TXT)

**S11 Data. Candidate spermine/spermidine synthase-encoding genes.**
(TXT)

**S12 Data. Alignment of the candidate spermine/spermidine synthase-encoding genes.**
(TXT)

**S13 Data.** (A) Z-score normalised environmental variables. (B) Correlation test *P*-value of the environmental variables before cleaning. (C) Spearman's correlation coefficient of the environmental variables before cleaning. (D) Correlation test *P*-value of the environmental variables after cleaning. (E) Spearman's correlation coefficient of the environmental variables after cleaning.
(XLSX)

## Acknowledgments

We would like to thank all colleagues from the *Tara* Oceans consortium as well as the *Tara* Ocean Foundation for their inspirational vision. This article is contribution number 140 of *Tara* Oceans.

## Author Contributions

**Conceptualization:** Charlotte Nef, Mohammed-Amin Madoui, Chris Bowler.

**Data curation:** Charlotte Nef, Éric Pelletier.

**Formal analysis:** Charlotte Nef.

**Funding acquisition:** Chris Bowler.

**Investigation:** Charlotte Nef.

**Methodology:** Mohammed-Amin Madoui, Éric Pelletier.

**Project administration:** Chris Bowler.

**Resources:** Éric Pelletier.

**Software:** Mohammed-Amin Madoui, Éric Pelletier.

**Supervision:** Mohammed-Amin Madoui, Éric Pelletier.

**Validation:** Charlotte Nef, Mohammed-Amin Madoui, Éric Pelletier.

**Visualization:** Charlotte Nef.

**Writing – original draft:** Charlotte Nef.

**Writing – review & editing:** Charlotte Nef, Mohammed-Amin Madoui, Éric Pelletier, Chris Bowler.

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
