## [Editor Report · Decision Letter 0]

7 Jul 2022

Dear Dr. Bowler, 

Thank you for submitting your manuscript entitled "Whole-genome scanning reveals selection mechanisms in epipelagic Chaetoceros diatom populations" for consideration as a Research Article by PLOS Biology.

Your manuscript has now been evaluated by the PLOS Biology editorial staff and I am writing to let you know that we would like to send your submission out for external peer review.

Once your full submission is complete, your paper will undergo a series of checks in preparation for peer review. After your manuscript has passed the checks it will be sent out for review. To provide the metadata for your submission, please Login to Editorial Manager (https://www.editorialmanager.com/pbiology) within two working days, i.e. by Jul 09 2022 11:59PM.

Kind regards,

Paula

Editor

PLOS Biology

---

## [Decision Letter · Decision Letter 1]

2 Sep 2022

Dear Dr. Bowler,

Thank you for your patience while your manuscript "Whole-genome scanning reveals selection mechanisms in epipelagic Chaetoceros diatom populations" went through peer-review at PLOS Biology. Your manuscript has now been evaluated by the PLOS Biology editors, an Academic Editor with relevant expertise, and by several independent reviewers.

In light of the reviews, which you will find at the end of this email, we are pleased to offer you the opportunity to address the comments from the reviewers in a revision that we anticipate should not take you very long. We will then assess your revised manuscript and your response to the reviewers' comments with our Academic Editor aiming to avoid further rounds of peer-review, although might need to consult with the reviewers, depending on the nature of the revisions.

**IMPORTANT - SUBMITTING YOUR REVISION**

*Resubmission Checklist*

*Published Peer Review*

*PLOS Data Policy*

*Blot and Gel Data Policy*

Sincerely,

Paula

---

Senior Editor

PLOS Biology

REVIEWS:

Reviewer #1: Marine microbes interactions.

Reviewer #2: Evolution of microplankton.

Reviewer #1: In the article, "Whole-genome scanning reveals selection mechanisms in epipelagic Chaetogeros diatom populations", Nef and coauthors use 11 Chaetoceros MAGs from the Tara oceans dataset and consider distribution and diversity, with a focus on how microdiversity relates to abiotic factors in the environment. This is an interesting study that aims to address quintessential questions in marine microbial ecology: how is diversity created and maintained in a fluid ecosystem with unrestricted connectivity between populations. As is always the case, these authors, like others, are limited by data availability. However, the manuscript is comprehensive and polished and marks a significant contribution to the field. I have a few general questions that I think, if clarified, would improve the manuscript, as well as some minor comments:

Given that Chaetoceros is described in the introduction as an abundant and globally distributed phytoplankton group with more than 239 species, is it surprising that only 11 Chaetoceros MAGs were recovered from the Tara dataset? I understand that the MAGs were not assembled for this particular publication, but it might be worthwhile to comment on this. Why is the number so low? How does it compare to other phytoplankton groups? 

I suggest including the recently published Chaetoceros genome mentioned in the discussion in Figure 1, so readers can quickly grasp how these MAGs compare to a genome from the same genus. It would also be helpful to provide more context regarding how closely related the model diatoms and Chaetoceros are: is there reason to suspect that genomes from these groups should have similar size, GC content, etc.? 

For the BUSCO results, please indicate how many BUSCOs were in the set (n=303?, the number included changes with version). How do BUSCO results turnout when you use the stramenopiles lineage instead of eukaryota? I would also recommend providing the more complete BUSCO results (Complete Single-copy, complete duplicated, fragmented, and missing) as these metrics may be especially important for evaluating MAGs. 

Regarding the phylogenetic reconstruction - I have trouble seeing that the MAGs ARC_189 and PSE_253 resolved at the level of C. dichaeta. Based on the tree in the figure, you could just as well say that they resolved at the level of Thalassiosira. Is it possible to construct another tree with more diatom references and a more closely related outgroup to better resolve which taxa these two MAGs are more closely related to? Rather than buscos, you could use single copy orthologs from an orthofinder (or similar) run and then build a tree with more concatenated genes. Alternatively, other methods may be used to better place these two MAGs, but as is, it is difficult for a reader to be confident that they are in the genus Chaetoceros.

Figure 3, B - it is not very clear what is being correlated here - relative abundance at each station? The caption could use a little more detail. Also, it appears that most of the pairs are negatively correlated, maybe -0.25? It would be helpful if the far end of the scale bar was labeled. Finally, Pearson's rho can only asses linearly related data, I would recommend using Spearman's correlation coefficient in this case.

I'm generally wondering about time frames for diatom evolution - is stratification a prolonged enough phenomenon to cause genetic differentiation, as is discussed regrading the station TARA_194? Is it also possible that genetic differentiation occurred in two different water masses that are temporarily interleaved? With only two depths and one time-point, it is difficult to make conclusions here?

Figure 4B - the values should be moved above the bars as they are difficult to read as is

Figure 4E - a map of the regions of the Arctic ocean included in the analysis would be helpful here

Examining the correlation between abiotic parameters and population structure - 

How were environmental factors normalized? Log scaled or z-scored? 

Were environmental parameters correlated? How did you test for multicolinearity? What was the the variance inflation factor of the model? 

The lack of statistical significance as determined by Mantel tests may reflect the number of parameters being tested. If only one variable from each group of covarying parameters is included, the test could have more power.

How was geographic distance computed? As the crow flies or based on oceanographic connectedness? Were currents taken into account? I suspect that it was as the crow flies, but I wonder if a more oceanographic distance would be more significant, particularly because it was mentioned earlier in the manuscript that longitude was more correlated with SNVs than latitude and this was attributed to strong oceanic currents in the arctic. 

Minor comment: In the introduction, I would support changing "superior trophic levels" to "higher trophic levels".

Reviewer #2: The article from Nef et al. "Whole-genome scanning reveals selection mechanisms in epipelagic Chaetoceros diatom populations" investigates 11 MAGs attributed to the diatom genus Chaetoceros recently published by Delmont et al. (2022). The article presents a glimpse of the probable future of environmental genomics: Using reference genome to extract MAGs from environmental datasets and investigate the ecology of targeted groups in the light of their molecular properties. 

I must say that although I have experience in metabarcoding, I have none in metagenomics and I am not well placed to criticize the technical part of the work because I never did it myself. I represent more the target audience from this type of study and I will comment it from this point of view.

The article reads well and I cannot do any critics on the content. I had some reservation when reading some results (e. g., Weak correlation between geography and population structure, non-significant mantel tests), but these points are addressed in the discussion. The text is generally dense, which makes sometimes difficult to absorb all the information, but I appreciate this type of reading and the sections are articulated logically. There are a few things that I would like to ask the author to add or clarify but this is only minor.

I did not understand why the authors worked with 11 MAGs "only". I know that it is a nice feat but I would like to know why there is not more (or less) genomes. Is it because of the coverage of the data? Or the authors limited themselves to the more dominant/complete MAGs? Maybe I missed the explanation but I feel the information is currently missing. If the information is provided in Delmont et al. (2022), I feel it should be provided in the present paper nonetheless. 

I feel the authors should supply a graphic support to introduce the Chaetoceros genus (Ideally Figure 1) to the audience. Specifically, I feel it would be useful to provide the occurrence of OTUs from the metabarcoding dataset from the TARA Oceans to show the biogeographic occurrence of Chaetoceros, together with the number of OTUs ascribed to this genus (Compared to the number of taxa in Algaebase). This will provide an appreciation of the difference of representation of the same genus between morphological, metabarcoding and metagenomics datasets. Also, I think that providing light microscopy images and eventually an SEM from Chaetoceros would be useful for the non-diatom experts, as well as a map with the a polar projection (like in the figure S14) next to classical projection to appreciate better the geographical organisation in the arctic. 

I would remove the mention of the FST from the abstract, as it is not yet defined and not every reader might know what it means.

I support the publication of this manuscript but as stated above, I cannot criticize to methodological part of the work and I hope this will be covered by other reviewers.

---

## [Editor Report · Decision Letter 2]

6 Oct 2022

Dear Dr. Bowler,

Thank you for your patience while we considered your revised manuscript "Whole-genome scanning reveals selection mechanisms in epipelagic Chaetoceros diatom populations" for publication as a Research Article at PLOS Biology. This revised version of your manuscript has been evaluated by the PLOS Biology editors and the Academic Editor.

Based on our Academic Editor's assessment of your revision, we are likely to accept this manuscript for publication, provided you satisfactorily address the following data and other policy-related requests.

1. DATA POLICY:

You may be aware of the PLOS Data Policy, which **requires that all data be made available without restriction**: http://journals.plos.org/plosbiology/s/data-availability. For more information, please also see this editorial: http://dx.doi.org/10.1371/journal.pbio.1001797

A) Supplementary files (e.g., excel). Please ensure that all data files are uploaded as 'Supporting Information' and are invariably referred to (in the manuscript, figure legends, and the Description field when uploading your files) using the following format verbatim: S1 Data, S2 Data, etc. Multiple panels of a single or even several figures can be included as multiple sheets in one excel file that is saved using exactly the following convention: S1_Data.xlsx (using an underscore).

B) Deposition in a publicly available repository. Please also provide the accession code or a reviewer link so that we may view your data before publication.

Regardless of the method selected, please ensure that you provide the individual numerical values that underlie the summary data displayed in the following figure panels as they are essential for readers to assess your analysis and to reproduce it: Figures 1, 2ABCDEF, 3AB, 4ABC, 5ABCDE, 6ABC, 7ABCDE, and Supplementary Figures S1AB, S2AB, S3, S4ABC, S5, S6AB, S7ABCDEFGHIJK, S8, S9, S10AB, S11, S12, S13, S14ABCD, S15, S16ABCD, S17ABC, S18, S19AB, S20, S21, S22, S24ABCD, S26ABCD, S27AB, S28AB.

**Please also ensure that figure legends in your manuscript include information on where the underlying data can be found, and ensure your supplemental data file/s has a legend.**

**Please ensure that your Data Statement in the submission system accurately describes where your data can be found.**

2. We suggest a change in the title: "Whole-genome scanning reveals environmental selection mechanisms that shape diversity in populations of the epipelagic diatom Chaetoceros".

We expect to receive your revised manuscript within two weeks.

*Published Peer Review History*

*Press*

Sincerely,

Paula

---

Senior Editor,

pjaureguionieva@plos.org,

PLOS Biology

---

## [Editor Report · Decision Letter 3]

27 Oct 2022

Dear Dr Bowler,

Thank you for the submission of your revised Research Article "Whole-genome scanning reveals environmental selection mechanisms that shape diversity in populations of the epipelagic diatom Chaetoceros" for publication in PLOS Biology. On behalf of my colleagues and the Academic Editor, Emiley Eloe-Fadrosh, I am pleased to say that we can in principle accept your manuscript for publication, provided you address any remaining formatting and reporting issues. These will be detailed in an email you should receive within 2-3 business days from our colleagues in the journal operations team; no action is required from you until then. Please note that we will not be able to formally accept your manuscript and schedule it for publication until you have completed any requested changes.

PRESS

Sincerely, 

Paula 

---

Senior Editor

PLOS Biology
